# The Mediterranean Ocean Colour Level 3 Operational Multi-Sensor Processing

Gianluca Volpe[1], Simone Colella[1], Vittorio E. Brando[1], Vega Forneris[1], Flavio La Padula[1], Annalisa Di Cicco[1], Michela Sammartino[1], Marco Bracaglia[1,2], Florinda Artuso[3], Rosalia Santoleri[1]

[1] Istituto di Scienze Marine, Via Fosso del Cavaliere 100, 00133, Roma, Italy
[2] Università degli Studi di Napoli Parthenope,Via Amm. F. Acton 38, 80133, Naples, Italy
[3] Agenzia nazionale per le nuove tecnologie, l'energia e lo sviluppo economico sostenibile, Dipartimento Ambiente, Centro Ricerche Frascati, Frascati, Italy

*Correspondence to*: Gianluca Volpe (gianluca.volpe@cnr.it)

**Abstract.** The Mediterranean near-real-time multi-sensor processing chain has been set up and is operational in the framework of the Copernicus Marine Environment Monitoring Service (CMEMS). This work describes the main steps operationally performed to enable single ocean colour sensors to enter the multi-sensor processing applied to the Mediterranean Sea by the Ocean Colour Thematic Assembly Centre within CMEMS. Here, the multi-sensor chain takes care of reducing the inter-sensor bias before data from different sensors are merged together. A basin-scale in situ bio-optical dataset is used both to fine-tune the algorithms for the retrieval of phytoplankton chlorophyll and attenuation coefficient of light, kd, and to assess the uncertainty associated with them. The satellite multi-sensor remote sensing Reflectance spectra better agree with the in situ observations than those of the single sensors. Here we demonstrate that the near-real time processing chain compares sufficiently well with the historical in situ datasets to be confidently used also for reprocessing the full data time series.

## 1    Introduction

The Copernicus Marine Environment Monitoring Service (CMEMS) is one of the six services of the Copernicus program. It provides regular and systematic reference information on the physical state, variability and dynamics of the ocean, ice and marine ecosystems for the global ocean and the European seas. CMEMS delivers both satellite and in-situ high-level products prepared by Thematic Assembly Centres (TACs) and modelling and data assimilation products prepared by Monitoring and Forecasting Centres (MFCs). The Ocean Colour Thematic Assembly Centre (OCTAC) builds and operates the European ocean colour operational service within CMEMS providing global, Pan-European and regional (Arctic Ocean, Atlantic Ocean, Baltic Sea, Black Sea, and Mediterranean Sea) ocean colour (OC) products based on earth observation from OC missions (Le Traon 2015, Von Schuckman 2017). The OCTAC bridges the gap between space agencies, providing ocean colour data, and all users who need the added-value information not available from space agencies. Presently, the OCTAC relies on current and legacy OC sensors: MERIS (MEdium Resolution Imaging Spectrometer) from ESA, SeaWiFS (Sea-

viewing Wide Field-of-view Sensor) and MODIS (Moderate Resolution Imaging Spectroradiometer) from NASA, VIIRS (Visible Infrared Imager Radiometer Suite) from NOAA, and most recently on the Copernicus Sentinel 3A OLCI (Ocean and Land Colour Instrument) sensor.

Starting from the Level-2 (L2) data downloaded from space agencies, the OCTAC generates Level-3 (L3) and Level-4 (L4) products in near-real time (NRT) and delayed time (DT) modes. Within CMEMS, L3 products refer to the single snapshot, or daily combined products, mapped onto a regular grid, while L4 are products for which a temporal averaging method and/or an interpolation procedure is applied to fill in missing data values. The NRT products are operationally produced daily to provide the best estimate of the ocean colour variables at the time of processing. These products are generated soon after the satellite swaths are available together with climatological ancillary data, e.g., meteorological and ozone data for atmospheric correction, and predicted attitude and ephemerides for data geolocation. In the DT processing, the updated ancillary data made available from the space agencies are used to improve the quality of the NRT data. NRT and DT data streams hence are designed to fulfil the operational oceanography specific requirements for near real time availability of high quality satellite data with a sufficiently dense space and time sampling (e.g., Le Traon et al., 2015). Generally, once a year, the full data time series undergoes a reprocessing (REP) to ensure most recent findings to be consistently applied and back propagated to all data. REP products are multi-year time series produced with a consolidated and consistent input dataset and processing software configuration, resulting in a dataset suitable for long-term analyses and climate studies (Von Schuckman et al., 2017, Sathyendranath et al., 2017 and references therein).

Within CMEMS, observations from multiple missions are processed together to ensure homogenized and inter-calibrated datasets for all essential ocean variables. Combining the observations from different platforms results in higher coverage as compared with those of the single sensors. Moreover, the multi-sensor product allows non-expert users to access a robust and less ambiguous source of information. Currently in the OCTAC, the NRT and DT multi-sensor L3 and L4 products are derived from MODIS-AQUA and NPP-VIIRS data, while REP includes observations from SeaWiFS, MODIS-AQUA, MERIS and NPP-VIIRS. Global REP products are derived from two datasets: the OC-CCI (Climate Change Initiative, www.esa-oceancolour-cci.org) funded by the European Space Agency and the Copernicus-GlobColour initially developed by Globcolour Project (www.globcolour.info) and then updated and produced in the framework of CMEMS. OLCI is foreseen to be included into the NRT/DT multi-sensor products in 2019 and in the REP when the quality of the data will be deemed suitable.

In general, DT and REP products are meant to answer different questions and to satisfy different needs such as assimilation into operational models and climate studies, respectively. As such, DT data are expected to be as accurate as timeliness allows. The accuracy of REP data need to be stable in time as these data, which are consistently processed with a single software version, are used for studying long time scale phenomena. For the sake of timeliness, the accuracy of the NRT-DT data is relaxed with respect to the one associated with REP time series. In this respect, one of the aims of this work is to propagate the REP configuration to the DT processing mode, allowing full compatibility between the two datasets and to extend the climate-fit-research to the most recent observations.

Regional products differ from their global counterparts as they are specifically derived to accurately reflect the bio-optical characteristics of each basin (e.g., Szeto et al., 2011; Volpe et al., 2007; Pitarch et al., 2016; D'Alimonte and Zibordi, 2003). Due to peculiarities in the optical properties, the Mediterranean Sea oligotrophic waters are less blue (30 %) and greener (15 %) than the global ocean (Volpe et al., 2007), causing an overestimation of the phytoplankton chlorophyll concentration (Chl) retrievals by standard global algorithms (e.g Bricaud et al., 2002, D'Ortenzio et al., 2002). In the last decade, more accurate regional bio-optical algorithms (e.g., MedOC4) were implemented in the single-sensor operational processing chains for the Mediterranean Sea (Santoleri et al., 2008; Volpe et al., 2012).

The main objective of this work is to provide Copernicus users with a comprehensive description of the method currently applied by GOS (the group for Global Ocean Satellite monitoring and marine ecosystem study, of the Italian National Research Council, CNR) in the OCTAC context of CMEMS to produce the L3 multi-sensor ocean colour product over the Mediterranean Sea. Next section (Data and Methods) describes the bio-optical dataset forming the basis for the development and validation of the regional algorithms for the Mediterranean Sea, an update of the MedOC4 parameterization, as well as the satellite data input and output of the operational processing chain. Section 3 gives an overview of the validation results obtained in the comparison between the multi-sensor satellite products and the in situ data.

## 2    Data and Methods

### 2.1    The Mediterranean Sea in situ bio-optical dataset: MedBiOp

The development of geophysical products that best reproduce the Mediterranean biogeochemical conditions relies on an in situ bio-optical dataset collected across the basin over twenty years (Figure 1). Several parameters are routinely measured both for general oceanographic purposes (e.g., water temperature, salinity, oxygen content, fluorescence and light attenuation) and for the calibration and validation of remote sensing data. These include phytoplankton pigment concentration via HPLC analysis (High Performance Liquid Chromatography), light absorption due to coloured dissolved organic matter (CDOM), light absorption due to algal and non algal particles as well as to total suspended matter (TSM), particulate back scattering and apparent optical properties such as remote sensing reflectance (Rrs) and the diffuse attenuation coefficient (Kd). In this work, the in situ Rrs dataset is used as input to update the MedOC4 Chl algorithm and to validate the multi-sensor satellite-derived Rrs product. The in situ Chl dataset is larger than the Rrs and all samples in correspondence of optical measurements are used to update the MedOC4 Chl algorithm, while all others are used to validate the multi-sensor satellite-derived Chl product. On the other hand, Kd measurements are only used to fine-tune the Mediterranean algorithm for ocean colour retrieval.

In the OC processing chain the primary parameters used to derive the geophysical products is the spectral Rrs. The most important objective of using the in situ radiometric measurements is to derive surface, above-water Rrs spectra from in-water profiles. The multispectral Satlantic profiling system (OCR-507) is made for measuring the upwelling radiance, $Lu(z,\lambda)$, the downward and the upward irradiance, $Ed(z,\lambda)$ and $Eu(z,\lambda)$, and includes a reference sensor for the downward irradiance,

Es(0,λ), mounted on the uppermost deck of the ship. A Sea-bird CTD and a tilt sensor are also part of the system. The radiometric measurements are acquired and processed following the method described in Zibordi et al. (2011). To increase the number of samples per unit depth, data are acquired using the multicast technique (D'Alimonte et al., 2010; Zibordi et al., 2004).

Multi-level data processing is achieved using the Software for the Elaboration of Radiometer Data Acquisitions (SERDA), developed at GOS. The processing steps follow the consolidated protocols for data reduction of in-water radiometry (Mueller and Austin, 1995; Zibordi et al., 2011). First, data are converted from digital counts into their physical units. A filter is applied to remove data with profiler tilt angle larger than 5°. In order to reduce the influence of the light variability during the measurements, data from each sensor are normalised by the above-water downwelling irradiance. A least-square

linear regression is performed on the log-transformed normalised data, whose slope determines the diffuse attenuation coefficients of spectral upwelling Radiance (Kl(λ)), spectral upwelling Irradiance (Ku(λ)) and spectral downwelling Irradiance (Kd (λ)); the exponents of the intercepts are the sub-surface quantities (Lu(0−, λ), Eu(0−, λ) and Ed(0−, λ)). Outliers due to wave perturbations are removed and identified in those points differing, by default, more than two standard deviations from the regression line. The depth layer normally considered as relevant for the extrapolation to the surface is

0.3-3 m, but can be changed on the basis of the characteristics of each profile. The upwelling sub-surface quantities (i.e. Lu(0−, λ), Eu(0−, λ)) are also corrected for the self-shading effect following Zibordi and Ferrari (1995) and Mueller and Austin (1995) using the ratio between diffuse and direct atmospheric irradiance, and the sea-water absorption. Using the primary sub-surface quantities, it is then possible to derive additional products such as the Q-factor at nadir (Qn(0⁻,λ)=Eu(0⁻,λ)/Lu(0⁻,λ)), the remote sensing reflectance (Rrs(λ)=0.543·Lu(0−,λ)/Es(0,λ)) or the normalized water-

leaving radiance (Lwn(λ)=Rrs(λ)·E0(λ) with E0(λ) being the extra-atmospheric solar irradiance; Thuillier et al., 2003). Fluorimetric measurements associated with CTD casts are used to increase the depth resolution of the HPLC-derived chlorophyll. These calibrated fluorimetric casts are then used to compute the optically weighted pigment concentration (OWP) as already reported in Volpe et al. (2007).

In addition to the MedBiOp dataset collected by GOS over the Mediterranean Sea, two fully independent datasets, collected

at fixed location, are included for the validation in this study: Rrs data estimated from above-water measurements at the Aqua Alta Oceanographic Tower (AAOT) as part of the AERONET-OC network in the northern Adriatic Sea (Zibordi et al., 2009), as well as Rrs data from the BOUSSOLE buoy located in the Ligurian Sea (Antoine et al., 2008; Valente et al., 2016). Moreover, for the validation of the diffuse attenuation coefficient we use the independent BGC-Argo float dataset from Organelli et al. (2016).

**2.2    Satellite Data Processing Chain**

As mentioned, GOS operates two different processing chains (Figure 2), for the NRT/DT and for the REP data production. The input of both processing chains is the spectral Rrs downloaded from upstream data providers. Hence, in both cases, the atmospheric correction is not part of these processing chains. This approach differs from the previous regional processing

chains which started from L1 (Volpe et al., 2007; 2012), as updates by the space agencies in the L1 to L2 processor resulted in a delay of months before it could be taken up in the operational processing chain.

As schematically shown in Figure 2, the NRT/DT chain consists of four parts aimed at populating a two-year rolling archive with multi-sensor Level-3 data at daily temporal resolution. The rolling archive includes the L3 obtained by the NRT L2 data (i.e., processed by with preliminary ancillary data, calibration known at the time of acquisition, preliminary climatology and so on), which are superseded, generally after one month, by the L3 produced in DT mode. Thus, the processing chain is exactly the same for the two modes, NRT and DT, what changes are the input data from space agencies. Data in the rolling archive are homogeneous in terms of format and processing software, meaning that if, for any reason, a change is made on the processing chain, the entire rolling archive is processed back for consistency. The ingested L2 data (R2018.0) currently derive from MODIS-AQUA and VIIRS sensors only. L2 are downloaded from the Ocean Biology Processing Group (OBPG) at NASA which use the l2gen processor for the atmospheric correction in its default parameterisation (Mobley et al., 2016). The NRT/DT chain involves the pre-processing of different sensors with different wavelengths (as detailed in Section 2.2.1) that are then merged together over a common set of wavelengths (Table 1, Section 2.2.2). Section 2.2.3 provides a description of the algorithms for the satellite-derived Chl estimation and for the attenuation coefficient of light at 490 nm (Kd490). As it will be detailed later, the inherent optical properties (IOPs: the absorption due to phytoplankton, $a_{ph}$, and to detrital and dissolved matter, $a_{dg}$, and the backscattering due to particles, $b_{bp}$, all at 443 nm) are used to align the different sensors over the common set of wavelengths. For this reason, the IOPs are an active part of the processing and are also made available to users as output of the chain.

For the REP processing, Rrs spectra over the common set of wavelengths (Table 1) are produced by the Plymouth Marine Laboratory (PML) using the OC-CCI processor version 3 (hereafter CCIv3, www.esa-oceancolour-cci.org) merging MERIS, MODIS-AQUA, SeaWiFS and VIIRS data. As fully detailed in CCI (2016a), SeaWiFS and VIIRS derive from the OBPG chain using the l2gen processor, while MERIS and MODIS-AQUA are processed with the POLYMER atmospheric correction processor (Steinmetz et al., 2011). At the moment of writing, the CCIv3 is based on the NASA reprocessing R2014.0. Within CMEMS, PML runs the regional CCIv3 processor at 1km resolution rather than at 4km as for the global OC-CCI dataset. In the following of this work, with CCIv3 we will refer to both the processor and the derived Rrs exclusively made for CMEMS, whereas with REP we will refer to the output of this chain, Chl and Kd490. These are consistently retrieved with the same algorithms as in the NRT/DT chain (Section 2.2.3), updated on a yearly basis and are available to users on the CMEMS web portal (marine.copernicus.eu).

As shown in Figure 1, most of the in situ data used for the validation analyses do not overlap with the two-year rolling archive (2017-2018, at the time of writing). Hence, for the sole scope of the product validation, the NRT/DT production chain was used to process the entire satellite data archive, including SeaWiFS and MERIS data. SeaWiFS data were obtained from NASA-OBPG (R2018.0), while MERIS data are the ESA third reprocessing with the POLYMER, made available by PML.

### 2.2.1 NRT/DT single sensor pre-processing

Once downloaded and quality checked, single-sensor L2 data are fed into the pre-processing chain to harmonize data from different sensors in terms of format, projection, and most of all in terms of a common set of wavelength bands. The quality checks that are operationally performed soon after the download are associated with the integrity of data files or their effective coverage over the region of interest (the Mediterranean Sea, in this case). Moreover, the pre-processing also takes care of sorting out issues that may affect one sensor only such as the destriping procedure or the removal of the bowtie effect.

*Destriping*

An important task, operationally performed over both MODIS-AQUA and NPP-VIIRS images is the application of a destriping procedure over L2 products to remove the instrument-induced stripes. These two sensors scan the Earth surface via a rotating mirror system that reflects the surface radiance to band detectors. Stripes originate from two hardware problems: i) the two sides of the mirror are not exactly identical, and ii) the band detector degradation is not homogeneous. Destriping correction is performed by applying the method developed by Bouali and Ignatov (2014) and adapted to ocean colour products by Mikelsons et al. (2014). The procedure splits the image into a stripe-affected and a stripe-free part. The stripe-affected part is then passed through a filter that removes the stripes, and then is added back to the stripe-free component to produce the final destriped image. The definition of striped and de-striped domains is achieved by measuring the gradients (both along and across the scan) and by selecting as "stripped" the ones below the pre-determined threshold values.

*Removal of the bowtie effect*

As sensor detectors have constant angular resolution, the sampled Earth area, i.e. the dimension of the pixel at ground, increases with the scan angle. This results in consecutive scans to overlap away from nadir, in turn giving the entire scan the shape of a bowtie. Differently than other sensors such as MODIS-AQUA, the aggregation scheme on board VIIRS removes this effect through a combination of aggregation and deletion of overlapping pixels, resulting in a series of rows of missing values at the edge of each L2 granule. These lines can be identified through the bowtie removal flag (BOWTIEDEL). In this production chain and in view of the sensor merging, these missing values are filled in by linear interpolation. Alas, the L2 flags associated with these pixels are not updated due to the difficulty of interpolating binary fields.

*Flagging & Mosaicking*

Each L2 granule is quality checked via the application of the L2 flags provided by Space Agencies. The L2 flags result from the atmospheric correction procedure and provide the sensing conditions at pixel scale. The flags currently applied are those of the OBPG standard processing (https://oceancolor.gsfc.nasa.gov/atbd/ocl2flags/), except for the atmospheric correction failure (ATMFAIL) flag that is not applied to VIIRS because it overlaps for almost all water pixels over the Mediterranean Sea with the BOWTIEDEL, thus effectively thwarting the interpolation of the lines affected by the bowtie effect. From a test over 645 granules (3200 x 3232 pixels each) acquired over the Mediterranean Sea in 100 days (10 April 2018 to 18 July

2018) it was found that only in 31 pixels the atmospheric correction failure flag was raised for pixels not affected by bow tie deletion or any of the other OBPG standard flags.

Moreover, each granule undergoes a further quality check by removing all isolated pixels (defined as those pixels with a meaningful value that are entirely surrounded by pixels with missing value) and by filling in all isolated missing pixels (defined as those pixels with the missing value that are entirely surrounded by pixels with a meaningful value) using the near-neighbourhood approach. All Rrs spectra are further checked for the presence of negative values, which may occur in the blue part of the spectrum due to the failure of the atmospheric correction; one negative value within the spectrum (excluding the 670nm band) is enough for the entire spectrum to be rejected. All available granules for each day are remapped at 1 km resolution on the Equirectangular grid covering the Mediterranean Sea (6°W-36.5°E, 30°N-46°N). All re-gridded granules from the same sensor and from the same day are mosaicked together into a single file containing the Remote Sensing Reflectance at nominal sensors' wavelengths.

*Band-shifting*

At the scale of the pixel, the goal is to merge single-sensor Rrs spectra into a single spectrum. The idea is that from the Rrs spectrum one can easily derive, directly or indirectly, all the geophysical parameters of interest not only for the ocean colour community, but also for the wider biogeochemical scientific community. One of the problems of the multi-sensor merging is the different set of bands of the various ocean colour sensors that have to be merged. Some bands are coincident (443 nm), others may differ by a few nanometres (486 and 488 or 410 and 412 nm) while others can be significantly different (e.g., the green bands of MODIS-Aqua, SeaWiFS and OLCI, which are 547nm, 555nm and 560 nm respectively, Table 1). A technique to collapse the various spectra on a pre-defined set of bands is thus essential for the multi-sensor merging; to this aim the band shifting method described by Mélin and Sclep (2015) was implemented here with the application of the Quasi Analytical Algorithm (QAA version 6, Lee et al., 2002 and following updates http://www.ioccg.org/groups/Software_OCA/QAA_v6_2014209.pdf) in forward and backward modes. Rrs is related to the absorption and scattering properties of the medium, which in turn are given by the additional contributions of all the medium components (seawater, particulate and dissolved matters). Starting from the Rrs at the sensor native wavelengths and from the characteristic spectral shapes of the IOPs, the QAA allows the estimation of the IOPs at target wavelengths. The QAA is then applied in forward mode to estimate the Rrs at these bands. The accuracy of QAA retrievals over the Mediterranean Sea was assessed with a limited number of observations by Pitarch et al (2016), who found that $b_{bp}$ at 555 nm was retrieved within 5% of in situ measurements across open and coastal waters. This approach produces a set of common bands (grey-shaded in Table 1) for all sensors and allows the daily merging of the Rrs from which it is then possible to apply algorithms to derive geophysical products. The uncertainty introduced by band shifting is estimated in most cases at well below 5% of the reflectance value (with averages of typically 1–2%), especially for open ocean regions (Mélin and Sclep, 2015).

### 2.2.2    NRT/DT multi sensor processing: Rrs spectra

Once single sensor spectra are homogeneous in terms of wavebands, it is possible for the Rrs from the available sensors to be merged together into single images. The output is a set of six Rrs images, each of which is treated as an individual image independently from the other Rrs bands of the spectrum.

*Differences between MODIS and VIIRS*

At pixel scale, several reasons can be at the base of the differences between two observations. The geometry of the observations constitutes an issue that is under the control of the atmospheric correction scheme. Since this part of the processing is performed by space agencies, this issue is rarely accounted for in the context of L3 multi-sensor merging, which instead only considers the radiometric quantities as fully normalized (Maritorena and Siegel, 2005). The order of magnitude difference between Rrs retrieved by MODIS and VIIRS varies with the wavelength (Figure S.1). The distribution of the Rrs ratio at 670 nm shows the most negative kurtosis. At 412 nm, the median Rrs ratio ranges between 0.7 and 1, while at 443 it improves and narrows to 0.85 and 1.05 with MODIS being in general below VIIRS. For the three other bands (490, 510 and 555 nm), the Rrs ratio distribution displays the narrowest spread around 1 with the median values ranging between 0.9 to 1.1. Moreover, a pixel is sampled with different geometry (scattering angle) and not exactly at the same time by the two sensors; in the Mediterranean Sea, the differences between the two sensor time overpasses do not exceed one hour. Here, we tested that the discrepancy between the two Rrs spectra cannot be ascribed to differences in the overpass times and/or to the geometry of the observation (Figure S.1). We argue that there must be other factors responsible of the observed differences such as inter-sensor calibration or even the various bands used for operating the single-sensor atmospheric correction (eliciting different responses by the atmospheric correction code and its assumptions/simplifications). All these issues should be addressed before any sensor merging can effectively be performed (Sathyendranath et al., 2017).

*Inter-Sensor Bias Correction*

Before merging all the available sensors together at any given time, their Rrs spectra are individually bias-corrected with respect to their references as detailed below. Here, we extend the method developed within OC-CCI for reducing the inter-sensor bias (CCI, 2016b), as this is a propedeutical step to the proper merging of data collected from different sensors. In practice, when two or more sensors are available for the same period, one sensor is taken as reference and the others are bias-corrected to the reference. For the inter-sensor bias to be corrected, daily climatological bias maps are computed at the same spatial resolution of the source data (e.g., 1 km). During the SeaWiFS era, the method is applied to SeaWiFS-MODIS-MERIS sensors having SeaWiFS as reference. From 2010 onward, the method is applied to the couple MODIS-VIIRS using MODIS as reference, after its bias with SeaWiFS is corrected. The climatological bias maps were computed using data from 2003 to 2007 for the SeaWiFS era, and from 2012 to 2014 for the other.

Briefly, the OC-CCI scheme to compute the daily climatological bias maps is:

1)    over the periods of reference, for each sensor, a rolling temporary daily average map of Rrs is computed (simple mean) over the period of 7 days: the data day itself plus 3 days before and 3 days after.

2) For each day, the ratio between the temporary average Rrs maps from the various sensors is computed.

3) This allows the calculation of 365 daily climatology maps of the ratio between each pair of missions over the periods of reference.

4) To increase map coverage and to reduce the spatial gradients, smoothing of the daily climatology bias maps over a temporal window of 2N+1 days (with N=60) are computed following equations 1 and 2:

$$\delta(d,x,y) = \frac{\sum_{i=-N}^{N} w_i \delta_i(d+i,x,y)\theta_i}{\sum_{i=-N}^{N} w_i \theta_i}, \tag{1}$$

with

$$w_i = \frac{N+1-|i|}{N+1}, \tag{2}$$

where $\delta(d,x,y)$ is the daily bias map climatology, and $\theta_i = 1$ if $\delta_i$ is associated with a valid value, zero otherwise. The value of the weight, $w$, decreases from 1, for the same day, to N/N+1 for the days before and after, to 1/N+1 for the first and last days of the ±N-day window.

The way the daily climatological bias maps are here computed differs from the OC-CCI technique. First, the rolling temporary seven-day average (point 1 of the OC-CCI method described above) is here computed using equations 1 and 2, with N=3. The smoothing of the daily climatology bias maps is obtained by applying a weighting-function (as point 4 of the OC-CCI method described above) in both space and time, contemporaneously. The spatial kernel of the 3x3 box centred to the pixel is defined as:

| | | |
|------|------|------|
| 0.25 | 0.50 | 0.25 |
| 0.50 | 1.00 | 0.50 |
| 0.25 | 0.50 | 0.25 |

The cumulative effect of these two weighting functions is given by their cross product.

Furthermore, the method was not applied to the 670 nm band because the percent difference between SeaWiFS and in situ observations at 670 nm is one or two orders of magnitude larger than the blue-green counterparts in both oligo- and meso-trophic conditions (MedBiOp, BOUSSOLE) (Section 3). Moreover, the number of matchups between SeaWiFS and all the available in situ data (MedBiOp, BOUSSOLE and AAOT) at 670 nm is ~40% of those in the blue-green spectral region (data not shown).

*Sensor-Merging*

When merging data from two or more sensors, three possible conditions can occur: i) the pixel is observed from more than one sensor, ii) the pixel is observed from one sensor only, iii) the pixel is in no clear sky condition or masked out because of any of the operational L2 flags, from all sensors. In the latter case the pixel is assigned the missing value. In the former two conditions the merging is not straightforward because it strongly depends on the ability to reduce the inter-sensor bias to zero. When the pixel is sampled by one sensor only, but the surrounding pixels by more than one or by the other sensors, there is an increasing probability of introducing artefacts or spatial gradients, which in reality do not exist and are only the

result of the merging procedure. To prevent the occurrence of such horizontal discontinuities, here we apply a smoothing procedure based on the use of the climatology field, described in Volpe et al. (2018) and summarized below. First, the field from each sensor (Figure 3a-b) is filled with the same relevant daily climatology (Figure 3e, see below for more details about the climatology), as shown in Figure 3c-d. Filling is performed as follows; for each sensor, the difference between the two

fields (observed and climatology) is first computed in correspondence of co-existing values. Such difference is propagated and smoothed all over the spatial domain. Missing observational values are replaced with the climatology corrected by the computed difference map. This prevents the generation of sharp gradients. At this stage, the simple average between all available climatology-filled sensor data is computed. Then all the non-clear sky pixels are set to the missing value (Figure 3f). This is the procedure operationally and currently applied to data acquired by MODIS-AQUA and VIIRS to produce the

multi-sensor Rrs product. It is important to note that features, which are only present in the climatology, but not in the daily single-sensor images, are also absent in the merged product. In the example of Figure 3, features of such a kind can be clearly identified in correspondence of the Strait of Bonifacio, in the Tyrrhenian Sea, which extends eastwards only in the climatology (Figure 3e) but in none of the other fields (MODIS-AQUA or VIIRS). Another example is given by the tongue of Modified Atlantic Water (Manzella et al., 1990) that penetrates the southern sector of the Sicily Channel towards the

Libyan coasts, which is present in AQUA, VIIRS, and in the merged image, but not in the climatology. Similarly, the Rhone River plume, visible in the climatology as a small reddish spot, is absent from both single-sensor images and from the merged multi-sensor product.

After all bands are merged, single pixel Rrs spectra are available (Figure 4) for the geophysical products to be computed. Within this step, a mask is computed for keeping track of the single sensor inputs to the multi-sensor product and added to

the NetCDF files (Figure 4b and Figure 4d). The examples show two cases of blue and greener waters along the Spanish coast and in the northern Adriatic Sea, respectively. In both cases, the bias correction demonstrates to improve the satellite Rrs estimate being closer to the in situ measurements, at all bands.

*Climatology*

As mentioned the climatology provides a spatial support to the sensor merging. The climatology field is obtained from the

thirteen years of SeaWiFS data. This daily field has the same spatial resolution (nominally 1 km at nadir) and projection (cylindrical) as the operational field. These climatology maps were created using the data falling into a moving temporal window of ± 5 days. Five days are deemed to be a good compromise between the need of filling the spatial domain and the de-correlation time scale of the OC data in the Mediterranean Sea; this has been estimated as being 3 days on average (the day at which the autocorrelation value halves, Volpe et al., 2018). The resulting daily climatology time series includes the

pixel-scale standard deviation, the average, the median, the modal, the minimum, and the maximum values. The next version of the NRT/DT processing chain will include a climatology field computed by taking into account the space-time weighted averaging and a longer and more recent data time series.

### 2.2.3    Level-3 geophysical products

The input to all algorithms used to derive the various geophysical products is the Rrs spectrum, which in this context derives from the NRT/DT processing chain described above and from the CCIv3 processor. It should be noted that in the L1 to L2 processing performed by the space agencies, the water leaving radiance normalization scheme makes use of Chl values estimated with standard algorithms. The differences between standard Chl and MedOC4 estimates in the Mediterranean Sea might affect the accuracy of the resulting Rrs. However, in the context of L3 multi-sensor merging this inconsistency cannot be accounted for without performing the L1 to L2 processing in house. The previous regional processing chains started from L1 and did take this effect into account (Volpe et al., 2007; 2012). On the other hand, in the operational oceanography framework, the need to keep the L2 to L3 processing chain readily up-to-date imposes a trade-off between accuracy and timeliness.

As shown in Figure 2, from this point on, the NRT/DT and the REP chains collapse as they both use the same algorithms for computing Chl, Kd and the IOPs. Next sections explain how the various algorithms are derived and applied to Rrs data for their operational application.

*Chlorophyll a concentration*

There are two main categories of Chl algorithms, empirical and semi-analytical. Even though the latter now show performance comparable to that of empirical algorithms, these still remain more robust and are generally preferred in the operational context (e.g., NASA processing). Recently, Sathyendranath et al. (2017) discussed the characteristics that remote sensing data must have to be used in climate studies. They pointed out that semi-analytical algorithms would be preferred to empirical ones because they do not rely on past observations, which are not necessarily the best approximation for future observations (Dierssen, 2010). However, they still lack the robustness, which is typical of the empirical family of algorithms (OReilly et al., 2000 among others).

Operational services such as CMEMS aim at providing data for a wide range of applications from the assimilation of open ocean observations into biogeochemical models (Teruzzi et al., 2014) to coastal monitoring programs (such as the Marine Strategy Framework Directive, e.g. Colella et al., 2016). Unfortunately, there is not yet a unique Chl algorithm able to perform with the same accuracy across different environments. For example open ocean waters are prevalently dominated by phytoplankton cells and their products of degradation; these waters are well represented by chlorophyll concentration and are generally referred to as Case I waters (Morel and Prieur, 1977). On the other hand, the optical properties of coastal regions are more often influenced by various water constituents not necessarily covarying with phytoplankton and are referred to as Case II waters. Since the offshore extension of the coastal waters may vary and be of several kilometres (pixels), depending on the sea and weather conditions (e.g., coastal filaments may extend several tens of kilometres in the open ocean), the adoption of static masks for the application of different algorithms would result in errors associated with the sharp fronts. One way to overcome this issue is to merge two Chl products into a single field, after the exact identification of the two realms (Mélin et al., 2011; Volpe et al 2012; Moore et al., 2014). At pixel scale, Rrs spectra are translated into Chl twice:

assuming the entire satellite scene to belong to Case I and to Case II waters, each with its own algorithm. Then, the water type identification follows the method developed by D'Alimonte et al. (2003) which uses the Mahalanobis distance between the satellite spectrum and the in situ reference spectra (in terms of the mean values and the covariance matrices of the two experimental datasets). For Case I and Case II waters the MedOC4 (Volpe et al., 2007) and CoASTS (Berthon et al., 2002,
Zibordi et al., 2002) datasets are used, respectively. This approach is one step towards the need of the scientific community of dealing with products performing equally well in both water types, or at least to know where the first ends and the second starts (Sathyendranath, 2011, OC-CCI user consultation). To address also the latter point evidenced by the OC-CCI user consultation, a water type mask resulting from the Case I-Case II merging step is conveniently stored into the NetCDF files and made available to users. Thus two different algorithms are used to derive Chl in the two optical domains: the ADOC4
algorithm (D'Alimonte and Zibordi, 2003) is used for the Case II domain, while algorithm for Case I constitutes the matter of the next paragraph.

*Mediterranean Sea – MedOC4 – Case I*

The algorithm used to retrieve Chl in the Case I waters of the Mediterranean Sea is an updated version of the MedOC4, a regionally parameterized Maximum Band Ratio (Volpe et al., 2007). Figure 5a shows both the regional and the global
algorithm (OC4v6, https://oceancolor.gsfc.nasa.gov/atbd/chlor_a/) functional forms superimposed to the in situ observations collected in the Mediterranean Sea. The Mediterranean Sea tends to be "greener" than the Pacific and Atlantic oceans for any Chl values due to higher CDOM concentrations (Volpe et al., 2007 and references therein). Considering that the empirical algorithms are the expression of the in situ data from which they are derived, this figure provides a means for understanding the need to regionalize the algorithms to avoid the significant Chl overestimation that would be obtained with the global
algorithm, as already fully documented in Volpe et al. (2007, and references therein).

An important point that has to be borne in mind is that the colour of the ocean, in terms of maximum band ratio (MBR), explains 74% of the entire phytoplankton variability, as expressed by the determination coefficient ($r^2$) between the chlorophyll concentration and MBR (Figure 5a). This points to the importance (more than 25%) of the second order variability of the ocean colour signal (Brown *et al.*, 2008) that should be accounted for by future versions of the operational
algorithms, in line with the recent recommendation about the use of ocean colour data for climate studies (Sathyendranath et al., 2017).

*Diffuse Attenuation Coefficient - Kd490*

Figure 5b (red dots) shows the in situ diffuse attenuation coefficient of light at 490 nm as a function of the Rrs ratio (R=$\log_{10}$(Rrs490/Rrs555)) collected in the Mediterranean Sea. Superimposed to the in situ dataset is also the algorithm
functional        form        (turquoise        line)        used        in        the        OBPG        processing        at        global        scale (https://oceancolor.gsfc.nasa.gov/atbd/kd_490/). It is clear that the global algorithm only marginally overlaps with the in situ data, thus prompting for a regional dedicated algorithm to be developed. Black line is the in situ data best fit computed as a fourth power polynomial expression of the Rrs ratio whose functional form is applied to the Multi Rrs ratio.

## 2.3    Validation framework

The validation of the satellite products was carried out by pairwise comparison with the in situ observations: Chl and apparent optical properties, e.g., kd490 and Rrs. For determining co-location between in situ and satellite data records all measurements acquired in the same day were used, as L3 data used in this study do not preserve the time information. Then, similarly to Zibordi et al. (2012), the median values is extracted from a 3 by 3 box centred on the in situ measurement coordinates, only if at least 5 pixels have valid values and the coefficient of variation is smaller than 20%. Bailey and Werdell (2006) use narrow time window for determining coincidence (i.e. no more than ± 3 h); Figure S.2 presents the percent difference between satellite and in situ Rrs for time windows ranging between ±1 and ±8 hours, assuming 10 am UTC as satellite overpass time. The range of variability of the relative difference is always within 1%, confirming recent results from Barnes et al. (2019).

The uncertainty associated with the in situ data is due to several factors, e.g., the sea conditions, the operator ability which in turn can introduce several contamination factors; hence, here we consider satellite and in situ observations to be both affected by uncertainties (Loew et al., 2017). Thus, for the matchup analysis, a type-2 regression (also called orthogonal regression) is implemented here (Laws and Archie, 1981). The statistical parameters for the assessment of satellite versus in situ data are listed in Table 2. For log-normally distributed variables (such as Chl and kd490) both datasets are log-transformed prior to computing the slope (S), the intercept (I) and the determination coefficient. A good match between the two observations is achieved when S is close to one and I is close to zero. The RMSD is the average distance of a data point from the fitted line, measured perpendicular to the regression line. RMSD and bias have the same units as the data from which they are derived.

## 3    Results and Discussion

This section provides the validation analysis for the operational NRT/DT retrievals of Rrs and Chl with the multi-sensor merged approach. The NRT/DT products (Multi) are available in the CMEMS catalogue as a rolling archive spanning two years, prior which REP products are available instead. As already mentioned, since most of the in situ data used for the validation analyses were collected earlier than 2017 (two years ago, at the time of writing), we used the NRT/DT production chain described in Section 2.2 to process the entire satellite data archive. The validation of the REP products based on the CCIv3 is also included for comparison.

*Temporal trend*

In this context and with the general aim of identifying any temporal dependence of the computed statistics, the analysis was made comparing the satellite products with space-time collocated in situ measurements for each campaign separately and for the whole dataset. No significant temporal behaviour emerged from the analysis (results not shown), highlighting that both in situ and satellite data are homogeneous in time and well calibrated. Similar results were recently yielded at global scale by Sathyendranath et al. (2017).

Figure 6 shows the relative difference between satellite and the MedBiOp Rrs spectra. Satellite Rrs at all available bands for each sensor are compared with the same in situ Rrs bands (Table 1). In general, the Rrs in the blue bands (443 and 490 nm) performs better than those at 412 nm or those in the green region (510 and 555 nm). As mentioned above, SeaWiFS, MODIS-AQUA and VIIRS are all processed with the l2gen processor, so that it is not surprising that these three sensors display a common spectral behavior with respect to in situ observations. On the other hand, MERIS is the only one exhibiting a positive difference with respect to in situ observations for the 412 and 443 bands. This is likely due to the different processing (performed by ESA) for the L1 to L2 processing of MERIS (see section 2.2 for details). Apart from the 670 nm band (76%), SeaWiFS performs generally better than the other sensors, thus supporting the choice of being selected as reference sensor for the blue-green spectral range. All other satellite data never exceed 15% relative difference when compared with in situ observations at basin scale (Table S.2 to Table S.7). A noticeable feature presented in Figure 6 is the wide variability of the computed statistics (given by the standard deviation bars) highlighting that the satellite data presented here do not substantially differ from the in situ observations. Table 3 shows the full statistics for the Multi Rrs product.

One of the main reasons for merging data from different sensors is to enhance the domain coverage by reducing the influence of both the cloud coverage and generally flagged or masked pixels as well as the out-of-satellite-swath areas; in all cases the use of a multi-sensor approach increases the probability of valid clear sky observations. Figure 7 shows the time series of the percent basin coverage for four single sensors (SeaWiFS, MERIS, MODIS-AQUA and VIIRS) and of the Multi product. The number of clear sky pixels of the Multi is on average larger than that of the single sensors by as much as 40% (Figure 7), with minimum impact during winter and maximum at summertime. The difference between periods of maxima and minima somehow reflects the cloud cover influence over the multi-sensor product, with the winter-time being characterized by both cloud-cover and out-of-satellite-swath, while the summer periods being mostly affected by out-of-satellite-swath masked areas. Moreover, the coverage is higher in the period 2002-2011 when SeaWiFS (until 2010), MODIS-AQUA and MERIS were operating simultaneously. Despite the loss in 2010 of SeaWiFS with its very wide swath, in the entire 2011 the gain (turquoise line in Figure 7) does not decrease substantially. After the loss of MERIS in 2012 the gain in the percent basin coverage dramatically decreases. Thus the basin coverage depends in first instance on the number of available sensors but also to the relationships of the orbital parameters among the various OC missions.

Results in Figure 6 are representative of the performances of the various satellite observations (both single- and multi-sensors) against the in situ measurements that were widely collected over the basin in the past twenty years, the MedBiOp dataset. Similarly, Figure 8 shows the comparison of the two multi-sensor time series (Multi and CCIv3) against three in situ datasets: the basin scale dataset (MedBiOp) and two fixed location datasets (section 2.1), one of which coastal (AAOT).

In general, one could expect the mismatch between satellite and in situ observations to be larger at the extreme bands of the spectrum, i.e., at 412 nm and 670 nm. In the first case, because of the spectral distance from the NIR bands used for the atmospheric correction, and in the second because of the generally very low Rrs values that pose a challenge for both in situ and satellite determination of the Rrs at this band. Here, the difference between satellite and in situ Rrs observations at these

extreme bands is of the same order of magnitude as the blue-green part of the spectrum when observed in open ocean (MedBiOp and BOUSSOLE) but not in the coastal domain (AAOT). CCIv3 and Multi Rrs present a general good agreement, differences being smaller than 5%; this low difference is likely due to the two source datasets which are derived from two different NASA reprocessings, R2014 and R2018; at 412 nm, and to a lesser extent at 443 nm, this difference is more pronounced (more than 5%) because the impact of the R2018 is larger at these bands. An even more evident difference (larger than 10%) is evident at 670 nm; here the impact of R2018 should be less important than in the blue bands. One important difference between Multi and CCIv3 is that Multi is not bias-corrected over SeaWiFS at this band (section 2.2.2); since SeaWiFS performances at this band are not as good as at the other bands it is reasonable to assume that this might be the cause of the observed discrepancy, further supporting the choice of not using SeaWiFS to bias-correct the other sensors in this band. Another important feature in Figure 8 is the general difference of satellite performance (both Multi and CCIv3) in coastal and open waters. Statistics associated with the two open ocean datasets (MedBiOp and BOUSSOLE) are of the same order of magnitude and much better than those computed in the coastal domain (AAOT).

Table 4 shows the number of matchups for each band of Multi and CCIv3 in correspondence of the three in situ datasets. Two aspects emerge: one linked to the difference between Multi and CCIv3 and the other between the 670 nm band and the other bands. As mentioned earlier, it is reasonable to assume that the different source data (R2018.0 for Multi and R2014.0 for CCIv3) is responsible for the differences in spatial coverage and hence in the number of matchups. Moreover, it should be mentioned that MODIS-AQUA used in the Multi processing chain derives from NASA R2018.0, while it derives from POLYMER atmospheric correction scheme for CCIv3. As for the differences between the 670 nm band and the other bands, the very noisy spatial patterns present in the daily images of the Rrs at 670 nm very often result, at the scale of the matchup pixels, in the coefficient of variation to exceed the 20% threshold (section 2.3).

Overall, despite their absolute differences, the two multi-sensor satellite products show a similar level of accuracy which suggests that the Multi processor is also suitable for the REP processing chain. This would provide the two benefits of reducing the number of upstream data provider and of giving the NRT/DT and REP products full compatibility.

_Matchup – Chl_

Figure 9 shows the matchups for the L3 Chl product for both processing modes, REP (derived from the CCIv3 Rrs) and NRT/DT (derived from the multi-sensor merged described in this study). To facilitate the comparison between the two satellite products, the matchup dataset includes only the points in which both satellite data are available. Both products are regularly distributed around the line of best agreement for the entire Chl range, although for in situ values larger than 0.3 mg m$^{-3}$ there is a noticeable dispersion increase. Table 5 shows the statistics of the four datasets plotted in Figure 9. To assess the uncertainties of the Multi Chl currently distributed on the CMEMS portal, the analysis was performed on the period in which VIIRS and MODIS co-exist, i.e. January 2012 onwards. Despite the different number of matchups (44 vs 710) and different Chl ranges (~0.04 – 2 vs ~0.007 – 9), statistics associated with the full time series are totally comparable with those obtained with the most recent data only (2012 to present) as denoted by the AV (MODIS-AQUA and VIIRS) subscript in both Figure 9 and Table 5.

To further assess the level of accuracy associated with the Chl retrieval from multi mission merged approach presented in this study, we compared with the results at global scale reported in Climate Assessment Report, CAR, (CCI, 2017). Differently than here, in the CAR, the Chl log-transformation was used to compute all the statistics, not only those associated with the linear fit (slope, intercept and determination coefficient, section 2.3). Therefore, for this analysis, we recomputed all the statistics of Table 5 accordingly (Table S.10). The in situ data used to compute the CAR statistics are much more numerous (14582, Table S.10). Nonetheless, results for the proposed regional algorithms as well as for CCI at global and Mediterranean scales show a general good agreement in terms of the determination coefficient, RMSD, CMRSD and the slope of the linear fit. The difference in the intercept only reflects the difference in the two dataset range of variability, with the global being wider and characterized by larger modal value (centred over ~1 mg m$^{-3}$, Figure 8 in CAR) than the MedBiOp (Figure 9).

*Matchup – kd490*

Figure 10a shows the validation result of the satellite-derived kd490 with respect to the in situ kd490 obtained from the BGC-Argo float dataset (Organelli et al., 2016). As a matter of comparison, both algorithms shown in section 2.2.3 and in Figure 5b are also presented. A general overestimation of the in situ kd490 is clearly visible in both the global and the Medkd algorithms for kd490 values below 0.04 m$^{-1}$ and 0.06 m$^{-1}$, respectively. Organelli et al. (2017) show analogous results with the satellite data overestimating the BGC-Argo – derived kd490 values below 0.1 m$^{-1}$ (their Figure 11). Here, since the satellite kd490 are derived from the two algorithms applied to the Multi Rrs, the level of accuracy that one can expect should be of the same order of magnitude as the one shown in Figure 6 and Table S.7. However, statistics associated with this matchup analysis show that both algorithms do perform worse than expected (Table S.9), especially the Medkd. There can be a series of reasons for which the two datasets (satellite and in situ) show good or bad agreement. One such a reason could be linked to the different space-time distribution of the calibration and validation datasets, but from the comparison of Figure 1 and Figure 10c it does not appear to be the case. The different data distribution can at least partially explain the poor performances of the Medkd with respect to the BGC-Argo kd490 (Figure 10b). As already mentioned in section 2.2.3, the Global algorithm appearing inadequate to represent the Mediterranean Sea conditions was superseded by the regional Medkd algorithm. The fact that in the validation exercise, the Global algorithm performs better than the Medkd is totally unexpected and, as suggested by Organelli et al. (2017), "*strongly warrants for further investigation*".

## 4    Conclusions

This work presented the latest achievements in the operational processing chain for ocean colour data stream for the Mediterranean Sea in the context of the European Copernicus Marine Environment Monitoring Service. The development of the multi-sensor merged product builds on the previous version of this chain, which was focused on parallel processing of single sensors (SeaWiFS, MODIS and MERIS, Volpe et al., 2012). The introduction of an operational multi-sensor merged product aims to meet the operational oceanography intrinsic requirement of "*One Question One Answer*". Three main steps

were implemented: band-shifting, inter-sensor bias correction, and the sensor merging. The band-shifting is implemented exactly as in Mélin and Sclep (2015), while the implementation of the inter-sensor bias correction differs from the OC-CCI technique (CCIb, 2016) in the temporal and spatial aggregation scales. The sensor-merging shown in this work is based on the use of the climatology as input to the smoothing procedure as described in Volpe et al. (2018). The output of this

processing chain is the Rrs spectrum that constitutes the input to all algorithms used to derive the various geophysical products. The Rrs computed with the multi-sensor merged approach shows good agreement when compared with in situ observations of the basin-scale MedBiOp dataset as well as the two fixed location AAOT and BOUSSOLE time series. Moreover, this work presents an updated version of the empirical algorithms for Chl and Kd retrievals for the Mediterranean Sea based on the extended MedBiOp dataset. The comparison with the in situ observations yields good results when applied

to both the Rrs derived from the CCIv3 processor and those derived from the multi-sensor merged processing shown here. This suggests the opportunity to use the proposed multi-sensor processing chain in the REP context as well.

## 5    Acknowledgements

We wish to thank the anonymous reviewers for detailed and pertinent comments that helped to greatly improve the manuscript. Giuseppe Zibordi, as PI of the AERONET-OC site of Venise, is warmly thanked for the Level-2 surface

reflectance data processing and site maintenance. The authors are also grateful to the BOUSSOLE project for maintaining and providing high-quality surface reflectance data used for the validation of the satellite observations. The International Argo Project (a pilot programme of the Global Ocean Observing System) and the national programmes that contribute to it (http://www.argo.ucsd.edu, http://argo.jcommops.org) are very much acknowledged. The NASA Ocean Biology Processing Group is strongly acknowledged for providing Level-2 data used as input to the processing chain. This work has been

performed in the context of the Ocean Colour Thematic Assembly Centre of Copernicus Marine Environment and Monitoring Service (Grant number: 77-CMEMS-TAC-OC-N).

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

## 7 Figures

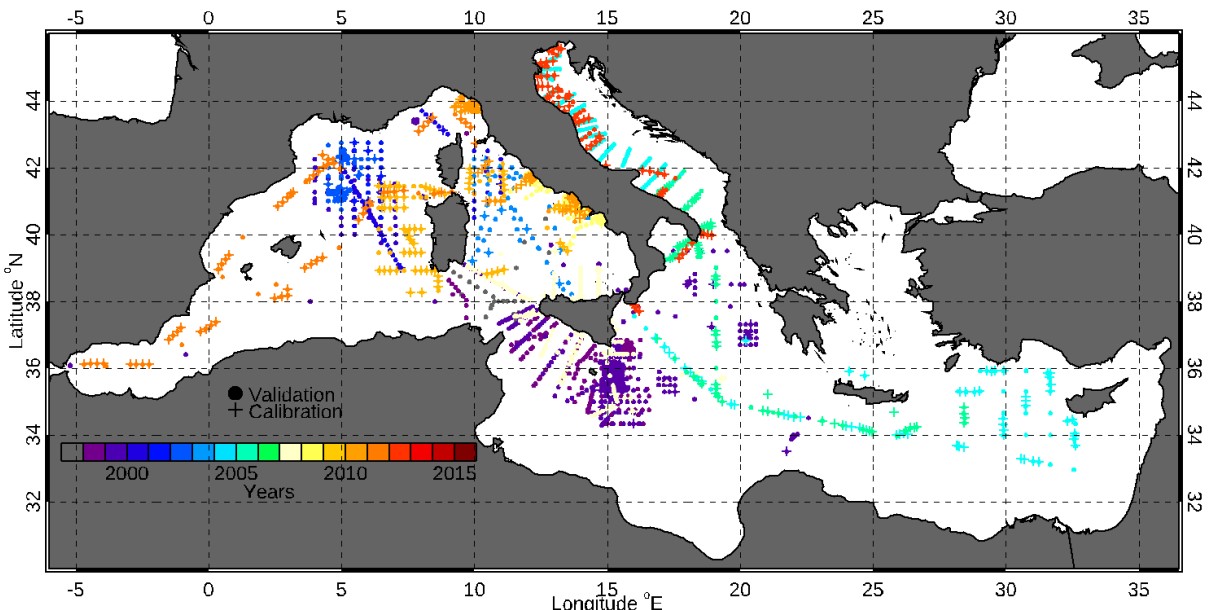

**Figure 1: Study Area and space-time distribution of the in situ MedBiOp dataset (1997-2016) used in this work. Dots identify the in situ station used as sea-truth for satellite data validation, whereas crosses refer to the observations used to develop the regional OC algorithms.**

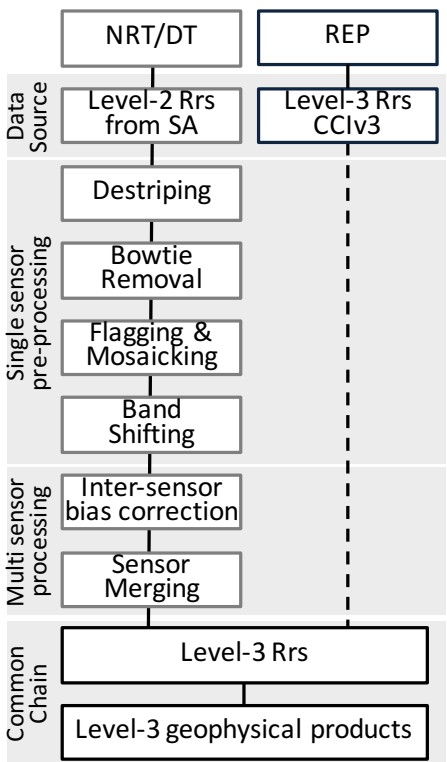

5    **Figure 2: flowchart of the processing chains for the two data production lines, NRT/DT and REP modes. SA stands for space agencies. The dashed vertical line indicates that, the CNR REP processing mode only involves the application of the regional fine-tuned algorithms for the retrieval of the geophysical quantities.**

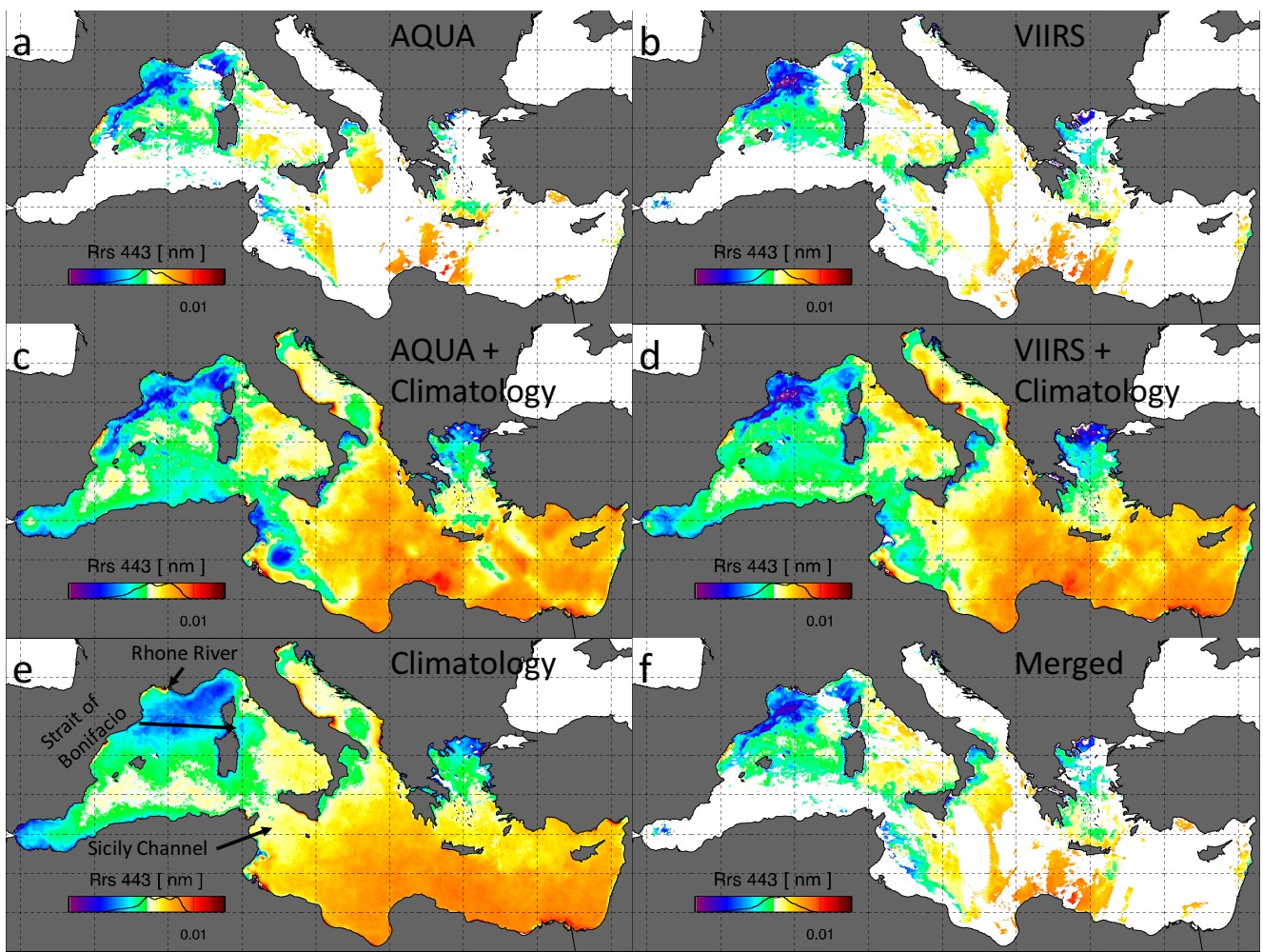

**Figure 3: Example of how the merging of MODIS and VIIRS works. Rrs 443 from MODIS AQUA (a), NPP-VIIRS (b) from April 1st 2012. Panels c and d are obtained by filling in panels a and b with daily climatology (e). The merged multi sensor product is obtained after removal of the unseen pixels (f).**

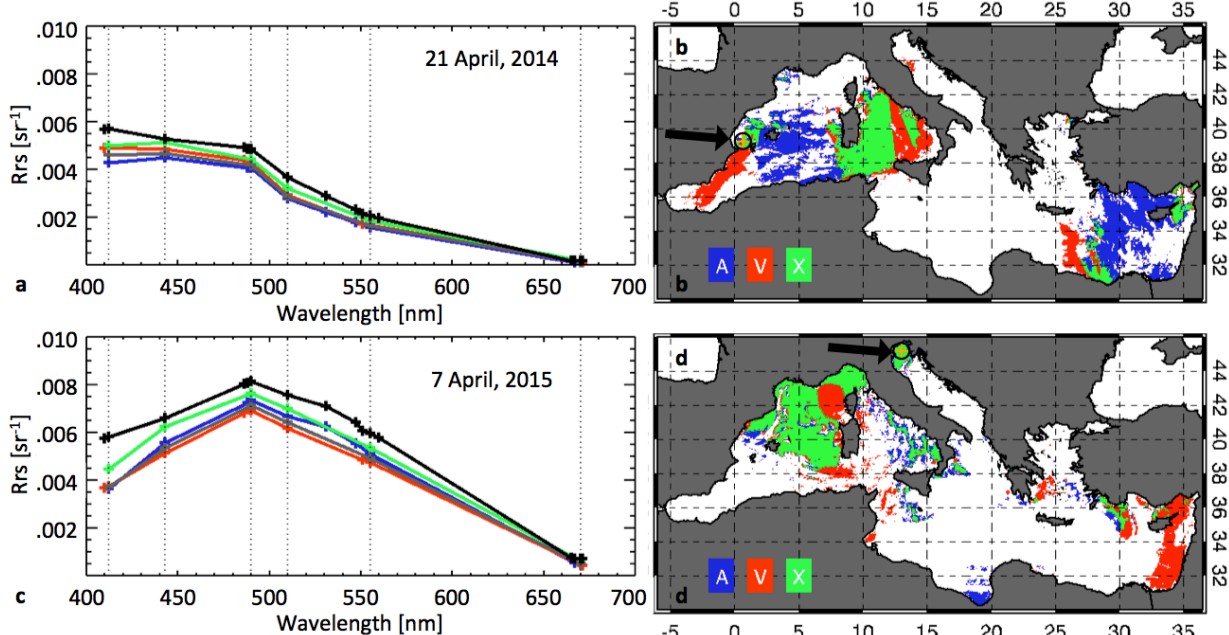

**Figure 4: Rrs spectra from the 21st April, 2014 (panel a), from MODIS-AQUA (A, blue), NPP-VIIRS (V, red), the merged multi-sensor product with the application of the bias correction (X, green) and without (grey), and the in situ measurements (black), all in correspondence of the in situ measurement location shown by the arrow in panel b. The map in panel b is the sensor mask of the day in which the pixels sampled by MODIS-AQUA only are shown in blue and those by NPP-VIIRS only in red; the pixels sampled by both sensors are shown in green. Panel c and d refer to the Rrs spectra and sensor mask from the 7th April 2015, in the northern Adriatic Sea.**

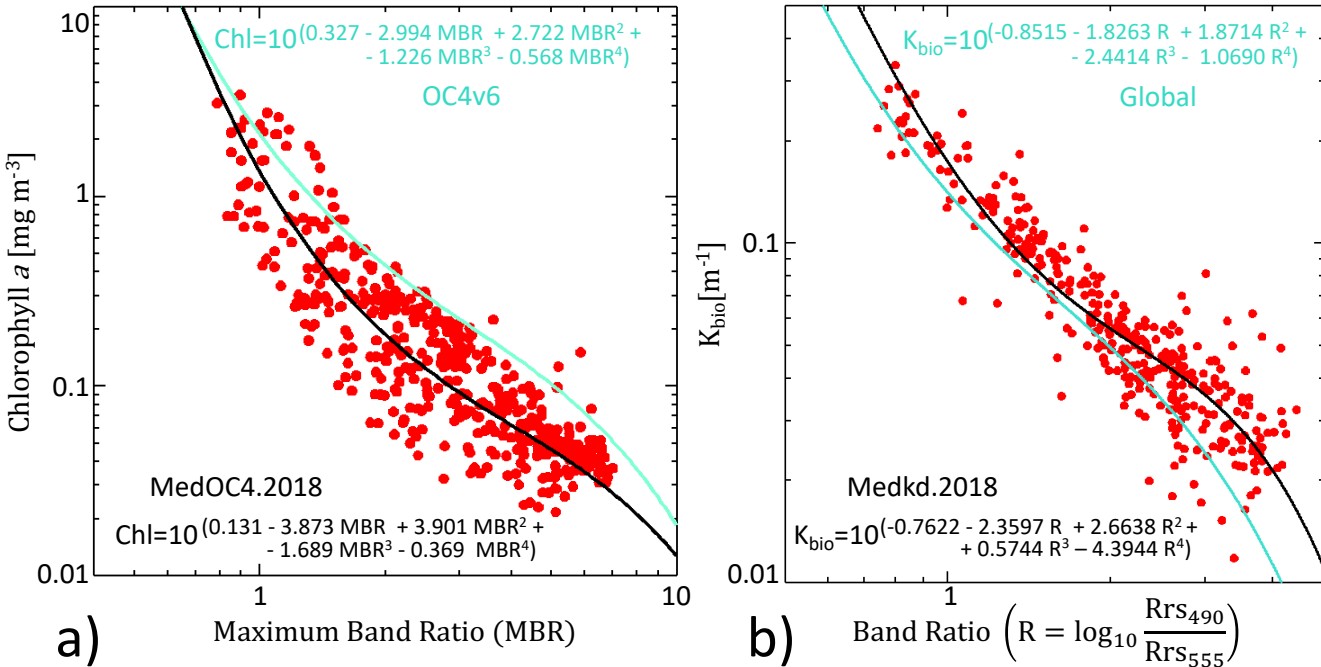

**Figure 5: panel a: algorithm for chlorophyll retrieval over the Mediterranean Sea. The maximum band ratio (MBR) is shown on the X-axis; it is the log$_{10}$ ratio between the maximum value between the three bands in the blue (443, 490 and 510 nm) and the one in the green part of the light spectrum (555 nm). Red dots (N=509) are the in situ bio-optical data (whose location is shown in Figure 1) used to compute the operational algorithm (black line). As a means of comparison the global algorithm (OC4v6, https://oceancolor.gsfc.nasa.gov/atbd/chlor_a) functional form is also superimposed (turquoise line). Panel b: algorithm for the retrieval of the diffuse attenuation coefficient, Kd490, over both the Mediterranean Sea (black line) and the global ocean (turquoise line). The global algorithm is the SeaWiFS (https://oceancolor.gsfc.nasa.gov/atbd/kd_490). Red dots (N=366) are the in situ measurements over the Mediterranean Sea. Kd490 is the sum of Kbio and of the attenuation due to pure sea water (0.0166; Mueller, 2000).**

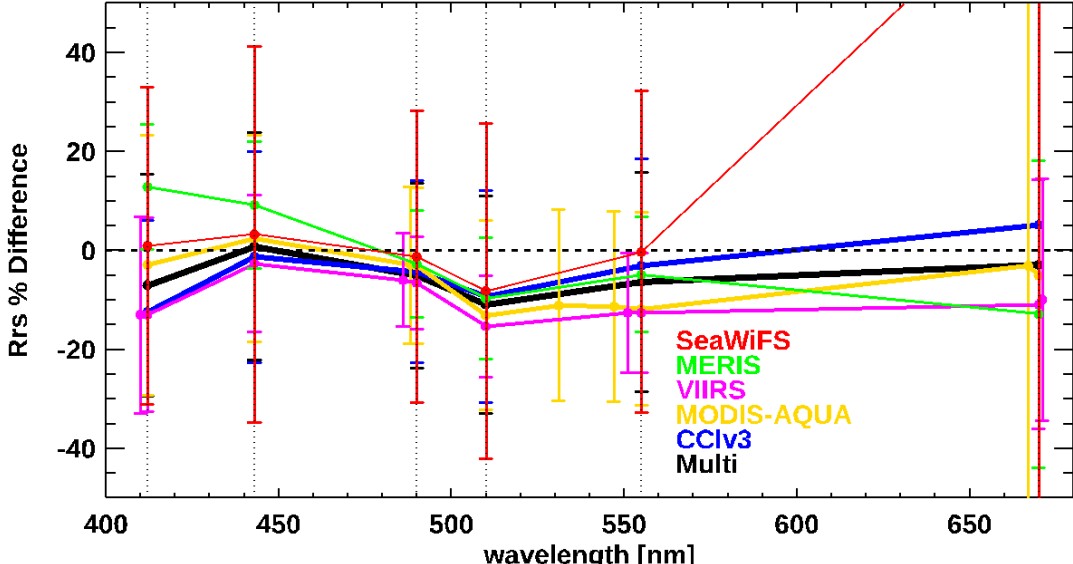

**Figure 6: Relative difference between satellite and MedBiOp Rrs spectra for MODIS-Aqua (yellow), NPP-VIIRS (magenta), SeaWiFS (red), MERIS (green), OC-CCI (blue) and for the multi-sensor product developed and described in this work (black). Vertical bars represent one standard deviation from the average RPD value. Target wavelengths are marked with vertical dotted lines.**

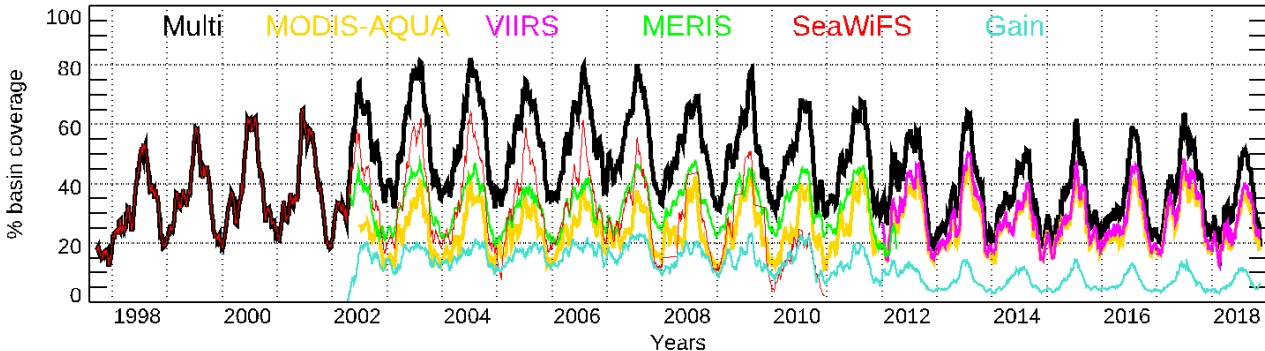

**Figure 7: Time series of the number of pixels for each satellite sensor as percent with respect to the basin total coverage. For the sake of readability, each line represents the result of the 30 days running median time series. Turquoise line is the basin coverage increase that is gained with the Multi with respect to the maximum coverage from the single sensors.**

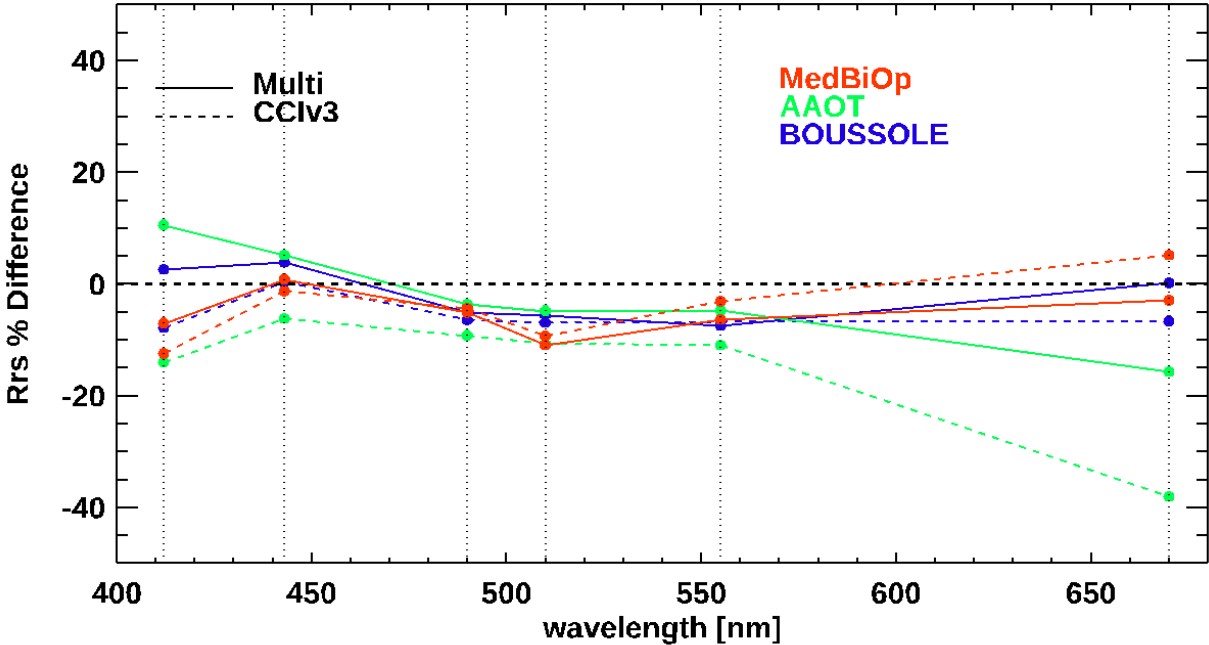

**Figure 8: Relative difference between Multi and CCIv3 satellite observations and in situ measurements (MedBiOp in red, AAOT in green and BOUSSOLE in blue). The number of matchups used from each dataset is summarized in Table 3. Target wavelengths are marked with vertical dotted lines. As a reference the two red lines correspond to the red and orange lines in Figure 6.**

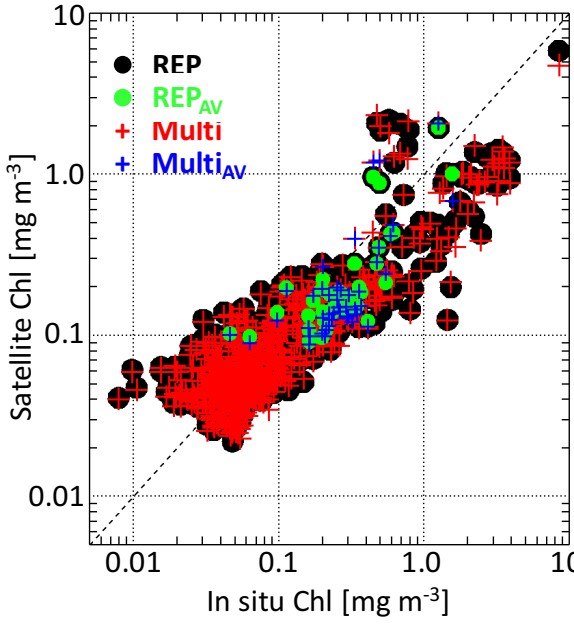

**Figure 9: Satellite (y axis) versus in situ (MedBiOp) Chl concentration. Satellite Chl is the REP (derived by the application of the MedOC4.2018 to the Rrs derived from the CCIv3 processor, black) and NRT (derived from the Multi processing, red). Green dots**

**and blue crosses are the REP and NRT for matchups on the period in which VIIRS and MODIS co-exist (REP_AV and Multi_AV). Statistics associated with the matchup comparison are shown in Table 4.**

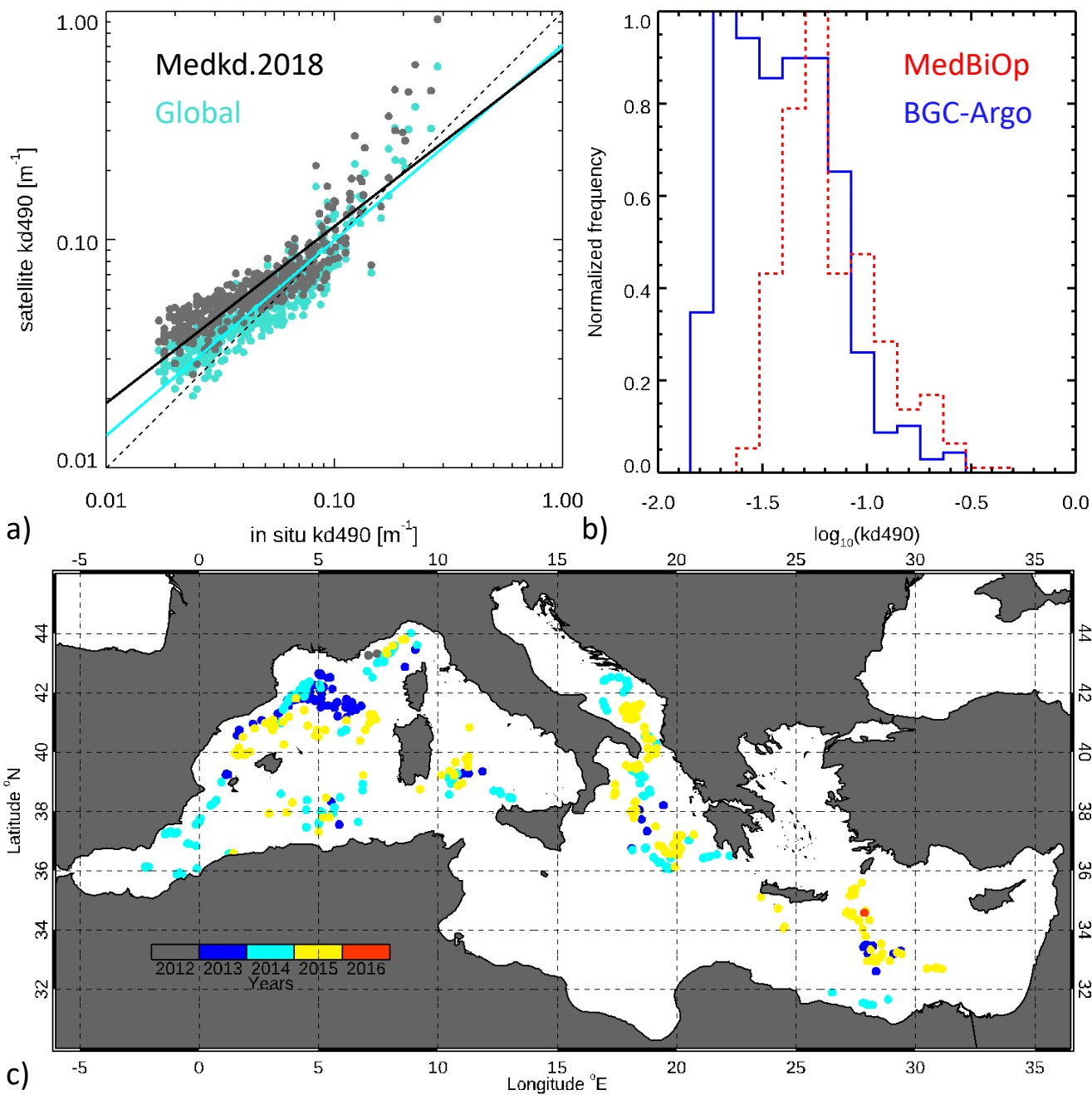

**Figure 10: Satellite kd validation with the BGC-Argo float dataset (Organelli et al., 2016). Panel a shows the in situ kd (x axis) versus the satellite-derived kd obtained with the Medkd.2018 (grey dots) and the Global algorithm (turquoise dots), respectively, as shown in Figure 5b. The two best-fit lines are also superimposed. The normalized frequency distribution of the two in situ**

## 8 Tables

|  | Sensors | | | | | | |
|---|---|---|---|---|---|---|---|
| Wavelength (nm) | VIIRS | MODIS | MERIS | OLCI | SeaWiFS | REP | In situ |
| 410 | • | | | | | | |
| 412 | | • | | | • | • | • |
| 413 | | | • | • | | | |
| 443 | • | • | • | • | • | • | • |
| 486 | • | | | | | | |
| 488 | | • | | | | | |
| 490 | | | • | • | • | • | • |
| 510 | | | • | • | • | • | • |
| 531 | | • | | | | | |
| 547 | | • | | | | | |
| 551 | • | | | | | | |
| 555 | | | | | • | • | • |
| 560 | | | • | • | | | |
| 665 | | | • | • | | | • |
| 667 | | • | | | | | |
| 670 | | | | | • | • | |
| 671 | • | | | | | | |

**Table 1: Overview of the available wavelengths from VIIRS, MODIS, MERIS, OLCI and SeaWiFS sensors and those used to produce the REP dataset (available from PML) and those collected in situ. Target wavelengths of the band shifting procedure are highlighted in grey. Column "in situ" refers to the bands of the Lu, Ed and Es Satlantic radiometers used to compute the algorithm functional forms and described in the in situ data description section (The Mediterranean Sea in situ bio-optical dataset: MedBiOp).**

| Name | Definition |
|---|---|
|  |  |

| | |
|---|---|
| Type-2 slope | $$S = \frac{\sum_{i=1}^{N}(X_i^E - \overline{X}^E)^2 - \sum_{i=1}^{N}(X_i^M - \overline{X}^M)^2 + \left[\left\{\sum_{i=1}^{N}(X_i^E - \overline{X}^E)^2 - \sum_{i=1}^{N}(X_i^M - \overline{X}^M)^2\right\}^2 + 4\left\{\sum_{i=k}^{N}(X_k^E - \overline{X}^E)(X_k^M - \overline{X}^M)\right\}^2\right]^{1/2}}{2\sum_{i=k}^{N}(X_k^E - \overline{X}^E)(X_k^M - \overline{X}^M)}$$ |
| Type-2 intercept | $$I = \overline{X}^E - S \cdot \overline{X}^M$$ |
| Determination coefficient | $$r^2 = \frac{\left[\sum_{i=1}^{N}(X_i^E - \overline{X}_i^E)(X_i^M - \overline{X}_i^M)\right]^2}{\sum_{i=1}^{N}(X_i^E - \overline{X}_i^E)^2 \sum_{i=1}^{N}(X_i^M - \overline{X}_i^M)^2}$$ |
| Root Mean Square Difference | $$\mathrm{RMSD} = \sqrt{\frac{\sum_{i=1}^{N}(X_i^E - X_i^M)^2}{N}}$$ |
| Bias | $$\mathrm{bias} = \frac{1}{N}\sum_{i=1}^{N}(X_i^E - X_i^M)$$ |
| Relative percentage Difference | $$\mathrm{RPD} = 100 \cdot \frac{1}{N}\sum_{i=1}^{N}\frac{X_i^E - X_i^M}{X_i^M}$$ |
| Absolute percentage Difference | $$\mathrm{APD} = 100 \cdot \frac{1}{N}\sum_{i=1}^{N}\frac{|X_i^E - X_i^M|}{X_i^M}$$ |

**Table 2: Metrics used to compare the estimated (satellite-based) dataset $X^E$ to a reference, measured in-situ dataset $X^M$. A more comprehensive table of metrics is provided in Supplementary Material (Table S.1).**

| Rrs | Slope | Intercept | $r^2$ | RMSD | Bias | RPD | APD | N |
|---|---|---|---|---|---|---|---|---|
| *412* | 0.99 | -0.0006 | 0.77 | 0.0015 | -0.00060 | -7 | 18 | 272 |
| *443* | 0.86 | 0.0007 | 0.73 | 0.0013 | -0.00023 | 1 | 15 | 272 |
| *490* | 0.65 | 0.0015 | 0.55 | 0.0013 | -0.00047 | -5 | 13 | 272 |
| *510* | 0.65 | 0.0009 | 0.57 | 0.0013 | -0.00060 | -11 | 18 | 272 |
| *555* | 0.68 | 0.0005 | 0.71 | 0.0012 | -0.00027 | -6 | 16 | 272 |
| *670* | 1.19 | -0.0001 | 0.91 | 0.0002 | -0.00002 | -3 | 35 | 197 |

**Table 3: Statistics associated with the Multi Rrs product computed over the MedBiOp dataset. The same statistics associated with**
5 **all products shown in Figure 6 are provided in Supplementary Material (Table S.2 to Table S.7).**

| | | Bands [nm] | | | | | |
|---|---|---|---|---|---|---|---|
| **in situ** | **Satellite** | **412** | **443** | **490** | **510** | **555** | **670** |
| **MedBiOp** | Multi | 272 | 272 | 272 | 272 | 272 | 197 |
| | CCIv3 | 262 | 262 | 262 | 262 | 262 | 223 |
| **AAOT** | Multi | 1794 | 1794 | 1794 | 1794 | 1794 | 1301 |
| | CCIv3 | 1753 | 1753 | 1753 | 1753 | 1753 | 1504 |
| **BOUSSOLE** | Multi | 961 | 961 | 961 | 961 | 961 | 780 |
| | CCIv3 | 882 | 882 | 882 | 882 | 882 | 780 |

**Table 4: Number of matchups used to compute the statistics presented in Figure 8.**

| Product | Slope | Intercept | $r^2$ | RMSD | Bias | RPD | APD | N |
|---|---|---|---|---|---|---|---|---|
| REP | 0.737 | -0.306 | 0.75 | 0.411 | -0.093 | 7 | 47 | 710 |
| Multi | 0.752 | -0.309 | 0.74 | 0.427 | -0.098 | 3 | 47 | 710 |
| $REP_{AV}$ | 1.052 | -0.108 | 0.57 | 0.207 | -0.064 | -18 | 43 | 44 |
| $Multi_{AV}$ | 1.184 | -0.047 | 0.50 | 0.271 | -0.057 | -17 | 48 | 44 |

**Table 5: Statistics about the three Chl matchup datasets described in Figure 9. The first two rows refer to the comparison of the two satellite multi-sensor products with the entire MedBiOp Chl dataset, while the last two refer to the statistics associated with matchups on the period in which VIIRS and MODIS co-exist ($REP_{AV}$ and $Multi_{AV}$). A more comprehensive table of metrics is provided in Supplementary Material (Table S.8).**