# Peer review of "The Mediterranean Ocean Colour Level 3 Operational Multi-Sensor Processing"

_Ocean Science, 2018_

## Referee Comment (RC1) · Anonymous Referee #1 · 5 Sep 2018

This is the review of the manuscript "The Mediterranean Ocean Colour Level-3 operational multi-sensor processing" by Volpe et al. The paper is mostly a description of the near-real-time/delayed-time processing of ocean color data in the framework of CMEMS. Results are mostly based on a validation analysis. Overall, the text reads more like a project report than a scientific paper, even if intended for a special issue on the European Copernicus Marine Service. It is actually incomplete or confusing in its description of various aspects of the processing (as shown by the list of secondary comments). So the text should be thoroughly re-organized to aim at clear descriptions (starting with the actual objectives of the paper) and the 'scientific' part should be reinforced before being considered for publication. These points are further developed below.

[Figure]

As demonstrated by the list of secondary comments, the description of the processing is often unclear and the pertinence of some processing steps is insufficiently supported. I understand that the authors ran into objective difficulties in documenting the processing chain with some elements that are fairly technical and apparently not fully described in literature (bow-tie effect, removing outliers, bias correction, smoothing. . .). I am wondering if some elements are described in more details in CMEMS (or other) reports / ATBDs that could be cited while simplifying the description of technical elements. An alternative is to use this paper as an opportunity to justify the choices made in the various processing steps and focus the work essentially on these. After the 'technical' part, the 'scientific' part is restricted to 2 pages (out of 13 pages of text) and could easily be reinforced with more discussion (currently there is very little description and discussion of the results, no comparison with published validation results, . . .). Based on the objectives of the paper, the authors should choose which part ('technical' or 'validation) to strengthen.

Besides the various points listed below, a more general lack of information can be noted when it comes to the comparison of processing chains in the validation analysis, CMEMS processing versus OC-CCI. The paper says that the CMEMS product used for validation is the near-real time (NRT) output. For me, this would mean that the data used in the validation analysis are those obtained in NRT mode, preserved as the processing went (i.e., computed with preliminary ancillary data, calibration at the date of acquisition, with climatology computed with very little data, etc. . .). In that case validation results reflect the quality of the data as downloaded NRT by users. Otherwise they are DT data, or even fully reprocessed data if they result from a consistent processing all the way through the time period (in terms of calibration, climatology computed with multiple years. . .). This (and implications) must be made clear in the manuscript (actually, how the validation results evolve as data are brought from NRT to DT and to reprocessed data is an interesting point). Besides the mode (NRT, DT) actually associated with the CMEMS data used in the analysis, more discussion should be given comparing those with the CCI data. The manuscript forgets to mention other

large differences between the CMEMS and CCI processing, including the atmospheric correction for certain sensors. In the extraction step, the grids of the products are also different (1-km versus 4-km if I am not mistaken). The study should clearly identify all possible sources of differences between the CMEMS and the CCI stream.

I have an issue with the field data. They are introduced in the study to explain the MedOC4 algorithm and the validation analysis. In such a context, a word on the uncertainties associated with the data would be necessary. Not clear to me is the distinction between the data serving for the development of MedOC4 and the data used for validation with match-ups (points of the former going into the latter are not fully independent validation points). As they are used for validation, BOUSSOLE and AERONET-OC data should be described a bit better. While I'm not familiar with the BOUSSOLE data distribution, there is a clear data policy for use of the AERONET-OC data (offer of authorship if I'm not mistaken) and I'm wondering if this has been respected (there is not even an acknowledgment in the manuscript).

Below are detailed comments, with requests for clarification/corrections and suggestions for improving the text. I'd recommend numbering the sections and sub-sections.

l.10: I'd suggest: "multi-sensor processing applied to the Mediterranean Sea by the Ocean Colour Thematic Assembly Centre of the Copernicus...": The abstract should be readable by readers who don't know about CMEMS, TAC, ...

l.11: "A basin-scale..."

l.12: "to fine-tune"

l.14: "than those"

l.15: "The Mediteranean...": information associated with this sentence should be relocated in the beginning of the abstract.

[Figure]

l.21: "CMEMS delivers. . ." rather than 'includes'

l.27: "users who"

l.28: please define all acronyms at first use.

l.2: "near-real time (NRT) and delayed time (DT) modes."

l.3: in OC jargon, monthly data computed from daily data are still termed L3.

l.15: is there a reference for this approach?

l.17-18: heavy sentence about 2 important benefits; should be reworded.

l.20: "are derived"

l.23: "is foreseen": so it is still not the case, which is at variance with the abstract.

l.26: "DT data"

l.26: "precise": what does this indicate exactly?

l.27: "both to be accurate": does it mean DT and REP? then the sentence should be restructured. What does 'consistent' mean here?

l.27: "For the sake of timeliness. . ."

l.4: "This work. . .": this paragraph describes the structure of the work, but the primary objective of the whole study is not clear. Page 2, line 24, "one of the aims" is mentioned, but how does it fit here, and what are the other aims?

l.14: "relies on an in situ. . ."

l.18: odd sentence; it means "absorption due to CDOM, absorption due to algal and non-algal particles, absorption due to TSM, and both AOPs and IOPs." absorption is

part of the IOPs, so which are the others measured?

l.20: are the in-situ IOPs used in this study?

l.30: are the authors sure about the choice of acronym . . . a large part of the scientific community is sufficiently well-versed in latin languages to understand the meaning of the word. . ..

l.2: "normalised by"

l.3: Kl and Ku should be defined.

l.9: "using the primary sub-surface quantities, it is then"

l.10: "such as the Q-factor"

l.18: "Chl": are Chl data from BOUSSOLE used in this work?

l.6: "using the OC-CCI. . ."

l.7: this sentence reads: "OC-CCI. . . at 1-km . . . rather than at 4-km for OC-CCI": unclear.

l.14: "Single-sensor pre-processing for NRT/DT modes" I presume.

l.15: "quality-checked": what does it entail?

l.15: I assume that the atmospheric correction applied in these cases is l2gen. This should be mentioned together with an appropriate reference.

l.22: "that reflects". . . "stripes originate"

l.32: "the dimension . . .": it happens also for the other sensors, doesn't it?

l.2: "these missing values": what is the benefit of this step?

l.6: and what about SeaWiFS and MERIS?

l.6: "space agencies": in that case it is only NASA.

l.7: "atmospheric correction failure": this is a strong step; I would assume that Rrs associated with that flag is just not usable as the AC failed according to the software. What is the criterion used by the software to consider the AC a 'failure'? what is then the status of Rrs? The authors should also support this decision with robust evidence that the resulting Rrs is actually valid.

l.8: "to avoid"

l.8: "salt and pepper": please define what this refers to.

l.9: "removing all isolated pixels...": in general this part needs clearer explanation. How is an "isolated pixel" defined?

l.13: please provide characteristics of the map (extent, resolution).

l.17: "spectra" from different missions?

l.21: "differ by"

l.25: "In general": may be removed.

l.30: "in theory": and in practice? Might be removed.

l.31: "apply algorithms to derive geophysical products"

l.3-5: not clear what the idea is here.

l.7: "Differences between MODIS and VIIRS"; and what about SeaWiFS and MERIS? In general some aspects of the manuscript associated with NRT/DT seem focused exclusively on MODIS/VIIRS (that has some logic since they are still active) but how

SeaWiFS and MERIS are handled should also be described as the related products are used in the validation analysis.

l.11: "normalized": does the BRDF correction make use of the OCI Chla value? In that, it is not consistent with the MedOC4 values. This should be acknowledged.

l.17: "we tested": by doing what?

l.19: "inter-calibration": it is certainly a factor but it is not the only one that could apply. I would still argue that geometry can play a significant role in the differences. Operating the atmospheric correction with different bands might also have an impact through the AC code (eliciting different responses by the AC code and its assumptions/simplifications).

l.22: it does not seem that the bias correction operated by CCI is described in that reference.

l.25: "climatological": what are the periods used for each sensor? is it the same?

l.28: "two steps": but then 3 are listed.

l.29: "temporary" ?

Page 8:

l.1: aren't the same equations used by CCI?

l.5: "weight w"

l.8: "and time": not necessary.

l.8: "as for OC-CCI": this should be written before. I think CCI also operates some type of spatial averaging.

l.12: "deemed insufficient": on the basis of what?

l.13: "40%": what is the reason for so many missing values at 670 nm? Negative

values? But earlier in the text, it is written that the quality check removes all incomplete spectra (if Rrs is <0 at any one band). So at that point, working with incomplete spectra for match-ups is inconsistent with the processing chain.

l.22: which "method"?

l.24: "are only the result of the merging procedure"

l.26-29: I have to say that I don't understand how the approach works. Volpe et al. (2018) not being published, the explanation should be clearer. For instance, is the climatology field computed as in the previous section? And then it is not clear how the smoothing operates.

l.2-8: The examples are not so obvious to me. I don't even see the Rhone plume on any map (should be a pattern leaving the coast. . .).

l.12: "In both cases": this is true for the cases shown but is it a general result? In any case, there is no reason to expect that the bias-corrected merged data would be closer to the field data than a simple average.

l.15: "The climatology field is obtained. . .": climatology is also mentioned in the 2 previous sections (bias correction, merging), so it is not clear what this SeaWiFS Chla climatology is for, nor why it is computed in a different way.

l.20: "this has been estimated": how? (or where?)

l.22: "To overcome. . .": but this type of filtering was already introduced for the Level-2 data. Do outliers appear again later in the processing ? Speaking of 'biases' here is not appropriate.

l.32:"Even though the latter now show performance comparable to that of empirical algorithms. . ."

[Figure]

l.1: "in discussing the characteristics that data must have to be used. . .."

l.2: "pointed out that . . ..": the sentence should be re-written.

l.2: "theoretical": to be replaced by 'semi-analytical' (what is a theoretical algorithm?)

l.10: "weather conditions"

l.15: "To identify. . . the identification. . .": sentence to be re-written.

l.17: "average spectra": the covariance matrix is also needed for such an approach.

l.21: "user survey": refers to the CCI user consultation?

l.6-24: in general I think this paragraph may be too long as the approach has already been well described in literature. Regardless of length I don't find it clear for a reader without a prior knowledge of the method.

l.12: "unable to perform"

l.15: "Inherent Optical Properties": I was therefore expecting validation results for the IOPs while there are none.

l.18: "in most instances": some example stats would be appropriate.

Page 12:

l.2: "used the NRT": meaning data processed as they were in NRT conditions (with preliminary ancillary data, calibration known at the time of acquisition, preliminary climatology, etc. . .)?

l.12: "common spectral behavior": is this a point that could be discussed?

l.18: "significantly": was any statistical test performed?

l.4: "significant"; what does it mean here?

l.5: "In the NIR": in the red?

Table 4: why are there less matchups at 670 nm (I understood that only complete spectra were kept)? The fact that the Multi and CCI products have a different number of match-ups should be discussed.

l.12: "NRT": or DT?

Figure 1: no so easy to distinguish validation from development points.

Figure 3: remind that the climatology fields are from a daily climatology.

Figure 5: "Pope and Fry 1972"?

Table 5: writes 'REP' versus 'Multi' while Table writes 'CCI' versus 'Multi'; coherence is needed to avoid confusion. While Table 4 shows different numbers of match-ups for the 2 products, here the numbers are the same. Is there an explanation for this?

---

## Referee Comment (RC2) · Anonymous Referee #2 · 27 Sep 2018

It is an excellent paper which explains in detail the scientific background behind the ocean colour products from CMEMS. I highly recommend for publishing. It will be an important scientific reference for the CMEMS OC-TAC services.

A few comments: 1. It is understood that the current product merging strategy follows the OC-CCI approach. Please explain whether moving to the Copernicus baseline of spectral band selection for the radiometry is planned in the future. 2. Future evolution to incorporate Copernicus missions, i.e. S3 OLCI (when ready), should also be brought up and discussed.

---

## Editor Comment (EC1) · P.-Y. Le Traon (Editor) · 12 Nov 2018

dear authors,

Please submit a revised version taking into account reviewer comments. Make sure all comments from reviewer 1 are taken into account. I will ask reviewer 1 to review the revised version.

best regards Pierre Yves Le Traon

---

## Author Response (AR1)

Dear Editor,

While editing for addressing all of the reviewers comments and suggestions we found that the quality of the manuscript could have been further improved by making a series of changes. These do not imply any significant change in the overall message but they still deserved to be addressed. This is the list of changes:

- The fields RMSD and Bias in Table 5 along with $\bar{X}^M$, $\bar{X}^E$, RMSD, bias CMRSD and MAE in Table S.8 all changed because in the current version we used the log-transformation only to compute the linear fit parameters (slope, intercept) and the determination coefficient. This is now explicitly mentioned in section 2.3 (validation framework).
- Figures 6, 7 and 8 and Tables 3, 4, 5 and Tables S.2 to S.8 changed in the current version because we realized that statistics were computed after the removal of the outliers without any mention in the manuscript. This had a minor impact on the statistics which were however corrected.
- MERIS time series has been added to Figure 7 along with the basin coverage increase that is gained with Multi product with respect to the best coverage obtained with the single sensors.
- Figure 5 was corrected to change the name of the Kd algorithm for the Mediterranean from MedOC4 to Medkd.2018. While editing we also recomputed that the Medkd.2018 coefficients (not the functional form).
- A new Table (S.10) has been added in Supplementary Material for the comparison with Chl matchup results from the Climate Assessment Report of the OC-CCI project.

Below is the point-by-point response to the reviews which include a list of all relevant changes made in the newer version of the manuscript. At the end of this file a marked-up manuscript version has also been appended.

Reply to Anonymous Referee #1

*We wish to thank Referee #1 for the very detailed and pertinent comments that helped to greatly improve the manuscript. We addressed all the general and specific comments as detailed below with a line-by-line response provided in italic.*

*In the revised version of the manuscript we will clearly identifying its objectives in line with the CMEMS special issue and the operational oceanography requirements. We will provide more details on the processing chain implementation to clarify several issues pointed out in this review. We will restructure section 2 on data and methods. We will evaluate the effects of the time window on the match-up analyses. We will improve the quality of Figure 1. We will add the validation of Kd using BGCArgo float data with a new figure in the body of the manuscript. We will add new figures in the supplementary materials.*

This is the review of the manuscript "The Mediterranean Ocean Colour Level-3 operational multi-sensor processing" by Volpe et al. The paper is mostly a description of the near-real-time/delayed-time processing of ocean color data in the framework of CMEMS. Results are mostly based on a validation analysis. Overall, the text reads more like a project report than a scientific paper, even if intended for a special issue on the European Copernicus Marine Service. It is actually incomplete or confusing in its description of various aspects of the processing (as shown by the list of secondary comments). So the text should be thoroughly reorganized to aim at clear descriptions (starting with the actual objectives of the paper) and the 'scientific' part should be reinforced before being considered for publication. These points are further developed below.

As demonstrated by the list of secondary comments, the description of the processing is often unclear and the pertinence of some processing steps is insufficiently supported. I understand that the authors ran into objective difficulties in documenting the processing chain with some elements that are fairly technical and apparently not fully described in literature (bow-tie effect, removing outliers, bias correction, smoothing. . .). I am wondering if some elements are described in more details in CMEMS (or other) reports / ATBDs that could be cited while simplifying the description of technical elements. An alternative is to use this paper as an opportunity to justify the choices made in the various processing steps and focus the work essentially on these. After the 'technical' part, the 'scientific' part is restricted to 2 pages (out of 13 pages of text) and could easily be reinforced with more discussion (currently there is very little description and discussion of the results, no comparison with published validation results, . . .). Based on the objectives of the paper, the authors should choose which part ('technical' or 'validation) to strengthen.

*In the new version of the paper we will highlight the nature of the work by clearly identifying its objectives in line with the CMEMS special issue and the operational oceanography requirements. We will provide more details on the processing chain implementation to clarify several issues pointed out in this review. We will evaluate the effects of the time window on the match-up analyses. We will add the validation of Kd using BGCArgo float data. We will add new figures in the supplementary materials.*

Besides the various points listed below, a more general lack of information can be noted when it comes to the comparison of processing chains in the validation analysis, CMEMS processing versus OC-CCI. The paper says that the CMEMS product used for validation is the near-real time (NRT) output. For me, this would mean that the data used in the validation analysis are those obtained in NRT mode, preserved as the processing went (i.e., computed with preliminary ancillary data, calibration at the date of acquisition, with climatology computed with very little data, etc. . .). In

that case validation results reflect the quality of the data as downloaded NRT by users. Otherwise they are DT data, or even fully reprocessed data if they result from a consistent processing all the way through the time period (in terms of calibration, climatology computed with multiple years. . .). This (and implications) must be made clear in the manuscript (actually, how the validation results evolve as data are brought from NRT to DT and to reprocessed data is an interesting point).

*The Reviewer is right. In the revised manuscript we will make clear that the matchup analysis refers to DT data. It would be very interesting to investigate on the quality drift that data may have between NRT and DT. However, since the CMEMS processing chain downloads the Level-2 from space agencies and this work involves the Level-1 to Level-2 processing step, this would require a reprocessing from Level-1 (using the NRT configuration) of all the days involved in the matchup analysis. The current chain is not meant for that.*

Besides the mode (NRT, DT) actually associated with the CMEMS data used in the analysis, more discussion should be given comparing those with the CCI data. The manuscript forgets to mention other large differences between the CMEMS and CCI processing, including the atmospheric correction for certain sensors. In the extraction step, the grids of the products are also different (1-km versus 4-km if I am not mistaken). The study should clearly identify all possible sources of differences between the CMEMS and the CCI stream.

*We will briefly describe the OC-CCI method with respect to the inter-sensor bias correction and mention the different atmospheric correction for certain sensors. The different spatial resolution was already mentioned in the original submission (page 5, line 7). However, we will rephrase the paragraph to make it more clear.*

I have an issue with the field data. They are introduced in the study to explain the MedOC4 algorithm and the validation analysis. In such a context, a word on the uncertainties associated with the data would be necessary. Not clear to me is the distinction between the data serving for the development of MedOC4 and the data used for validation with match-ups (points of the former going into the latter are not fully independent validation points). As they are used for validation, BOUSSOLE and AERONET-OC data should be described a bit better. While I'm not familiar with the BOUSSOLE data distribution, there is a clear data policy for use of the AERONET-OC data (offer of authorship if I'm not mistaken) and I'm wondering if this has been respected (there is not even an acknowledgment in the manuscript).

*Figure 1 showing the Cal/Val dataset will be redrawn to better show their distribution, and from which it will appear clear that the calibration data are not being used for validation. The reviewer is right; the section acknowledgement was missing. We will fill it acknowledging AERONET-OC, BOUSSOLE, BGC-Argo and NASA OBPG as those providing high quality data and which we are grateful to. The AERONET-OC PI declined our offer of co- authorship .*

Below are detailed comments, with requests for clarification/corrections and suggestions for improving the text. I'd recommend numbering the sections and sub-sections.

*We will number all sections and sub-sections.*

l.10: I'd suggest: "multi-sensor processing applied to the Mediterranean Sea by the Ocean Colour Thematic Assembly Centre of the Copernicus. . .": The abstract should be readable by readers who don't know about CMEMS, TAC, . . .

*OK*

l.11: "A basin-scale. . ."

*OK*

l.12: "to fine-tune"

*OK*

l.14: "than those"

*OK*

l.15: "The Mediteranean. . .": information associated with this sentence should be relocated in the beginning of the abstract.

*OK*

l.21: "CMEMS delivers. . ." rather than 'includes'

*OK*

l.27: "users who"

*OK*

l.28: please define all acronyms at first use.

*OK*

l.2: "near-real time (NRT) and delayed time (DT) modes."

*OK*

l.3: in OC jargon, monthly data computed from daily data are still termed L3.

*We will add the terms "Within CMEMS" at the beginning of the sentence to stress that this is the current terminology adopted by the Copernicus service.*

l.15: is there a reference for this approach?

*We will remove the statement from the introduction and add sentence framing this approach in the conclusions.*

l.17-18: heavy sentence about 2 important benefits; should be reworded.

*We will rephrase and split the sentence.*

l.20: "are derived"

*OK*

l.23: "is foreseen": so it is still not the case, which is at variance with the abstract.

*This sentence refers to OLCI, which is still not included in the multi-sensor product. It will probably be so by 2019. We do not see any discrepancy with what is stated into the Abstract, where OLCI is not mentioned.*

l.26: "DT data"

*OK*

l.26: "precise": what does this indicate exactly? l.27: "both to be accurate": does it mean DT and REP? then the sentence should be restructured. What does 'consistent' mean here?

*The sentence will be rephrased into "As such, DT data are expected to be as accurate as timeliness allows. The accuracy of REP data need to be stable in time as these data are consistently processed with a single software version."*

l.27: "For the sake of timeliness. . ."

*OK*

l.4: "This work. . .": this paragraph describes the structure of the work, but the primary objective of the whole study is not clear. Page 2, line 28, "one of the aims" is mentioned, but how does it fit here, and what are the other aims?

*We will add a sentence explicitly stating that the overall objective of the work is to provide Copernicus users with a comprehensive description of the method currently applied in the OCTAC context of CMEMS to produce the multi-sensor ocean colour product over the Mediterranean Sea. Propagating the REP configuration to the DT processing mode is part of the method.*

l.14: "relies on an in situ. . ."

*OK*

l.18: odd sentence; it means "absorption due to CDOM, absorption due to algal and non-algal particles, absorption due to TSM, and both AOPs and IOPs." absorption is part of the IOPs, so which are the others measured? l.20: are the in-situ IOPs used in this study?

*We will rephrase the sentence to avoid confusion and to more clearly specify what are the in situ data that were actually used in this study.*

l.30: are the authors sure about the choice of acronym . . . a large part of the scientific community is sufficiently well-versed in latin languages to understand the meaning of the word.

*We will change to "Multi-level data processing is achieved using the Software for the Elaboration of Radiometer Data Acquisitions (SERDA)".*

l.2: "normalised by"

*OK*

l.3: Kl and Ku should be defined.

*OK*

l.9: "using the primary sub-surface quantities, it is then"

*OK*

l.10: "such as the Q-factor"

*OK*

l.18: "Chl": are Chl data from BOUSSOLE used in this work?

*Actually they are not. We will remove Chl from the sentence.*

l.6: "using the OC-CCI. . ."

*OK*

l.7: this sentence reads: "OC-CCI... at 1-km ... rather than at 4-km for OC-CCI": unclear.

*We will rephrase the entire paragraph.*

l.14: "Single-sensor pre-processing for NRT/DT modes" I presume.

*It will be changed to "NRT/DT single-sensor pre-processing"*

l.15: "quality-checked": what does it entail?

*We will add a sentence to briefly explain the kind of data quality checks that are operationally performed.*

l.15: I assume that the atmospheric correction applied in these cases is l2gen. This should be mentioned together with an appropriate reference.

*We will add a sentence in the previous paragraph to mention that the NRT/DT data are downloaded from the OBPG at NASA that uses the l2gen in its default parameterisation for the L1 to L2 processing. Later in the paragraph we will add similar details for the REP processing.*

l.22: "that reflects". . . "stripes originate"

*OK*

l.32: "the dimension . . .": it happens also for the other sensors, doesn't it?

*Yes. We will rephrase the paragraph to make it clear.*

l.2: "these missing values": what is the benefit of this step?

*We will specify that the benefit is particularly evident in view of the sensor merging.*

l.6: and what about SeaWiFS and MERIS?

*To generalize we removed the reference to MODIS-AQUA and VIIRS. Please see also response to comment P7.L7*

l.6: "space agencies": in that case it is only NASA.

*OK*

l.7: "atmospheric correction failure": this is a strong step; I would assume that Rrs associated with that flag is just not usable as the AC failed according to the software. What is the criterion used by the software to consider the AC a 'failure'? what is then the status of Rrs? The authors should also support this decision with robust evidence that the resulting Rrs is actually valid.

*We will add details on the processing of pixels identified through the bowtie removal flag. We will clarify that the atmospheric correction failure flag is not applied to VIIRS because it overlaps with the bowtie removal flag for almost all water pixels. We tested the 645 granules (each of 3200x3232 pixels) acquired over the Mediterranean Sea in 100 days (10 April 2018 to 18 July 2018) and found that only in 31 pixels the atmospheric correction failure flag was raised for pixels not affected by bow tie deletion or any of the other OBPG standard flags.*

l.8: "to avoid"

*OK*

l.8: "salt and pepper": please define what this refers to.

*OK*

l.9: "removing all isolated pixels...": in general, this part needs clearer explanation. How is an "isolated pixel" defined?

*We will add a sentence to specify the meaning of the isolated pixel.*

l.13: please provide characteristics of the map (extent, resolution).

*OK*

l.17: "spectra" from different missions?

*Yes. We actually wrote "to merge single-sensor Rrs spectra into a single spectrum". Adding from "different missions" would be redundant.*

l.21: "differ by"

*OK*

l.25: "In general": may be removed.

*OK*

l.30: "in theory": and in practice? Might be removed.

*OK*

l.31: "apply algorithms to derive geophysical products"

*OK*

l.3-5: not clear what the idea is here.

*We will remove this sentence.*

l.7: "Differences between MODIS and VIIRS"; and what about SeaWiFS and MERIS? In general some aspects of the manuscript associated with NRT/DT seem focused exclusively on MODIS/VIIRS (that has some logic since they are still active) but how SeaWiFS and MERIS are handled should also be described as the related products are used in the validation analysis.

*We will add a sentence in section 2.2 (satellite data processing chain) to clarify that the current NRT/DT processing only involves MODIS-AQUA and VIIRS platforms, and that for the sole scope of the product validation we used the Multi-sensor production chain described in that section (2.2) to process the entire satellite data archive from 1997, thus including SeaWiFS and MERIS data.*

l.11: "normalized": does the BRDF correction make use of the OCI Chla value? In that, it is not consistent with the MedOC4 values. This should be acknowledged.

*We will add a sentence acknowledging the issue in the result section when commenting the Rrs matchup analysis, focusing on the trade-offs between accuracy and timeliness in the operational oceanography framework. Furthermore, this issue will be also the topic of a new paragraph discussing the the accuracy of Rrs in section 4.*

l.17: "we tested": by doing what? l.19: "inter-calibration": it is certainly a factor but it is not the only one that could apply. I would still argue that geometry can play a significant role in the differences. Operating the atmospheric correction with different bands might also have an impact through the AC code (eliciting different responses by the AC code and its assumptions/simplifications).

*At pixel scale, the cosine of the scattering angle was compared with the ratio between MODIS-AQUA and VIIRS Rrs for each band. As it will be shown by the new figure in Supplementary Material, no specific pattern was evident through this analysis. We will acknowledge that there might be some other factors linked to the AC influencing the difference between sensors.*

l.22: it does not seem that the bias correction operated by CCI is described in that reference.

*OK, reference will be corrected. This paragraph will be partially rewritten to address the series of comments below about the implemented bias-correction method.*

> l.25: "climatological": what are the periods used for each sensor? is it the same? l.28: "two steps": but then 3 are listed. l.29: "temporary" ? Page 8: l.1: aren't the same equations used by CCI? l.5: "weight w" l.8: "and time": not necessary. l.8: "as for OC-CCI": this should be written before. I think CCI also operates some type of spatial averaging.

l.12: "deemed insufficient": on the basis of what?

*One of the output of Figure 6 is that the performances of SeaWiFS at 670 nm were the worst as compared with the other bands and sensors. This justifies the choice of not using this sensor band as reference for bias adjusting the other sensors. This sentence will be rephrased.*

l.13: "40%": what is the reason for so many missing values at 670 nm? Negative values? But earlier in the text, it is written that the quality check removes all incomplete spectra (if Rrs is <0 at any one band). So at that point, working with incomplete spectra for match-ups is inconsistent with the processing chain.

*In the "Flagging & Mosaicking" paragraph, we will correct "one negative value within the spectrum (excluding the NIR bands) is enough for the entire spectrum to be rejected" replacing NIR with 670nm.*

l.22: which "method"?

*We will replace "method" with "merging".*

l.24: "are only the result of the merging procedure"

*OK*

l.26-29: I have to say that I don't understand how the approach works. Volpe et al. (2018) not being published, the explanation should be clearer. For instance, is the climatology field computed as in the previous section? And then it is not clear how the smoothing operates.

*We will rephrase the sentence into "To prevent the occurrence of such horizontal discontinuities, here we apply the smoothing procedure described in Volpe et al. (2018) and based on the use of the climatology field, described below". Volpe et al. (2018) is now published. We will also add a few sentences to better explain how the smoothing operates.*

l.2-8: The examples are not so obvious to me. I don't even see the Rhone plume on any map (should be a pattern leaving the coast. . .).

*We will add labels on Figure 3e to easy the reading.*

l.12: "In both cases": this is true for the cases shown but is it a general result? In any case, there is no reason to expect that the bias-corrected merged data would be closer to the field data than a simple average.

*The validation results of Figure 6 show that this is a general result.*

l.15: "The climatology field is obtained. . .": climatology is also mentioned in the 2 previous sections (bias correction, merging), so it is not clear what this SeaWiFS Chla climatology is for, nor why it is computed in a different way.

*We will make a clear difference between the daily climatology bias map used in the context of the bias correction and the field climatology used for the sensor merging. We will also remove the incorrect bit "using the MedOC4 regional algorithm for CHL (Volpe et al., 2007)". We will add a sentence to mention that the next version of the processing chain will include a climatology field*

*computed following the method adopted in the bias correction section.*

l.20: "this has been estimated": how? (or where?)

*We will add the appropriate reference.*

l.22: "To overcome. . .": but this type of filtering was already introduced for the Level-2 data. Do outliers appear again later in the processing ? Speaking of 'biases' here is not appropriate.

*Since it is mentioned earlier in the manuscript, we will remove this part from this context.*

l.32:"Even though the latter now show performance comparable to that of empirical algorithms. . ."

*OK*

l.1: "in discussing the characteristics that data must have to be used. . .."

*OK*

l.2: "pointed out that . . ..": the sentence should be re-written.

*We will rephrase the sentence*

l.2: "theoretical": to be replaced by 'semi-analytical' (what is a theoretical algorithm?)

*OK*

l.10: "weather conditions"

*OK*

l.15: "To identify. . . the identification. . .": sentence to be re-written.

*We will rephrase the sentence.*

l.17: "average spectra": the covariance matrix is also needed for such an approach.

*We will add the necessary details to the sentence.*

l.21: "user survey": refers to the CCI user consultation?

*Yes, we will change it.*

l.6-24: in general I think this paragraph may be too long as the approach has already been well described in literature. Regardless of length I don't find it clear for a reader without a prior knowledge of the method.

*We hope that after the suggested changes the paragraph will read better.*

l.12: "unable to perform"

*We will add the results of validation analysis performed using the BGC-Argo float L3 data (Organelli et al., 2017, Earth Syst. Sci. Data).*

l.15: "Inherent Optical Properties": I was therefore expecting validation results for the IOPs while there are none.

*We will remove the IOPs from this sentence. A reference to the Pitarch et 2016 al for the assessment of QAA based bbp in the the Mediterranean Sea will be added in the methods section at 2.2.1 where the use of QAA is first described.*

l.18: "in most instances": some example stats would be appropriate.

*We will rephrase this sentence to show that the overall validation results are not significantly affected by the temporal window. We will add a new figure in Supplementary Material to show the statistics behaviour with a variable temporal window.*

Page 12:

l.2: "used the NRT": meaning data processed as they were in NRT conditions (with preliminary ancillary data, calibration known at the time of acquisition, preliminary climatology, etc. . .)?

*We will correct NRT into NRT/DT. As mentioned earlier, we will add a couple of sentences in Section 2.2 (Satellite Data Processing Chain) where we better explain the differences between NRT and DT data and that for the sole scope of the product validation we used the NRT/DT production chain described in that section (2.2) to process the entire satellite data archive from 1997, thus including SeaWiFS and MERIS data.*

l.12: "common spectral behaviour": is this a point that could be discussed?

*We will add some discussion on this point.*

l.18: "significantly": was any statistical test performed?

*No, we did not perform any statistical tests. However, "do not significantly differ" here referred to the fact that the standard deviation bars in Figure 6 do overlap to one another. We will rephrase the statement by replacing "significantly" with "substantially".*

l.4: "significant"; what does it mean here?

*We will change "significant" to "relevant".*

l.5: "In the NIR": in the red?

*OK*

Table 4: why are there less matchups at 670 nm (I understood that only complete spectra were kept)? The fact that the Multi and CCI products have a different number of match-ups should be discussed.

*In the revised version of the manuscript we will expand on this.*

l.12: "NRT": or DT?

*We will correct to NRT/DT.*

Figure 1: not so easy to distinguish validation from development points.

*We will make it better*

Figure 3: remind that the climatology fields are from a daily climatology.

*OK*

Figure 5: "Pope and Fry 1972"?

*Corrected to Mueller (2000).*

Table 5: writes 'REP' versus 'Multi' while Table writes 'CCI' versus 'Multi'; coherence is needed to avoid confusion. While Table 4 shows different numbers of match-ups for the 2 products, here the numbers are the same. Is there an explanation for this?

*A thorough check throughout the manuscript will ensure consistency of the use of REP to identify the dataset and CCIv3 to identify the source processing chain for Rrs.*

Reply to Anonymous Referee #1

*We wish to thank Referee #1 for the very detailed and pertinent comments that helped to greatly improve the manuscript. We addressed all the general and specific comments as detailed below with a line-by-line response provided in italic.*

*In the revised version of the manuscript we will clearly identifying its objectives in line with the CMEMS special issue and the operational oceanography requirements. We will provide more details on the processing chain implementation to clarify several issues pointed out in this review. We will restructure section 2 on data and methods. We will evaluate the effects of the time window on the match-up analyses. We will improve the quality of Figure 1. We will add the validation of Kd using BGCArgo float data with a new figure in the body of the manuscript. We will add new figures in the supplementary materials.*

This is the review of the manuscript "The Mediterranean Ocean Colour Level-3 operational multi-sensor processing" by Volpe et al. The paper is mostly a description of the near-real-time/delayed-time processing of ocean color data in the framework of CMEMS. Results are mostly based on a validation analysis. Overall, the text reads more like a project report than a scientific paper, even if intended for a special issue on the European Copernicus Marine Service. It is actually incomplete or confusing in its description of various aspects of the processing (as shown by the list of secondary comments). So the text should be thoroughly reorganized to aim at clear descriptions (starting with the actual objectives of the paper) and the 'scientific' part should be reinforced before being considered for publication. These points are further developed below.

As demonstrated by the list of secondary comments, the description of the processing is often unclear and the pertinence of some processing steps is insufficiently supported. I understand that the authors ran into objective difficulties in documenting the processing chain with some elements that are fairly technical and apparently not fully described in literature (bow-tie effect, removing outliers, bias correction, smoothing. . .). I am wondering if some elements are described in more details in CMEMS (or other) reports / ATBDs that could be cited while simplifying the description of technical elements. An alternative is to use this paper as an opportunity to justify the choices made in the various processing steps and focus the work essentially on these. After the 'technical' part, the 'scientific' part is restricted to 2 pages (out of 13 pages of text) and could easily be reinforced with more discussion (currently there is very little description and discussion of the results, no comparison with published validation results, . . .). Based on the objectives of the paper, the authors should choose which part ('technical' or 'validation) to strengthen.

*In the new version of the paper we will highlight the nature of the work by clearly identifying its objectives in line with the CMEMS special issue and the operational oceanography requirements. We will provide more details on the processing chain implementation to clarify several issues pointed out in this review. We will evaluate the effects of the time window on the match-up analyses. We will add the validation of Kd using BGCArgo float data. We will add new figures in the supplementary materials.*

Besides the various points listed below, a more general lack of information can be noted when it comes to the comparison of processing chains in the validation analysis, CMEMS processing versus OC-CCI. The paper says that the CMEMS product used for validation is the near-real time (NRT) output. For me, this would mean that the data used in the validation analysis are those obtained in NRT mode, preserved as the processing went (i.e., computed with preliminary ancillary data, calibration at the date of acquisition, with climatology computed with very little data, etc. . .). In

that case validation results reflect the quality of the data as downloaded NRT by users. Otherwise they are DT data, or even fully reprocessed data if they result from a consistent processing all the way through the time period (in terms of calibration, climatology computed with multiple years. . .). This (and implications) must be made clear in the manuscript (actually, how the validation results evolve as data are brought from NRT to DT and to reprocessed data is an interesting point).

*The Reviewer is right. In the revised manuscript we will make clear that the matchup analysis refers to DT data. It would be very interesting to investigate on the quality drift that data may have between NRT and DT. However, since the CMEMS processing chain downloads the Level-2 from space agencies and this work involves the Level-1 to Level-2 processing step, this would require a reprocessing from Level-1 (using the NRT configuration) of all the days involved in the matchup analysis. The current chain is not meant for that.*

Besides the mode (NRT, DT) actually associated with the CMEMS data used in the analysis, more discussion should be given comparing those with the CCI data. The manuscript forgets to mention other large differences between the CMEMS and CCI processing, including the atmospheric correction for certain sensors. In the extraction step, the grids of the products are also different (1-km versus 4-km if I am not mistaken). The study should clearly identify all possible sources of differences between the CMEMS and the CCI stream.

*We will briefly describe the OC-CCI method with respect to the inter-sensor bias correction and mention the different atmospheric correction for certain sensors. The different spatial resolution was already mentioned in the original submission (page 5, line 7). However, we will rephrase the paragraph to make it more clear.*

I have an issue with the field data. They are introduced in the study to explain the MedOC4 algorithm and the validation analysis. In such a context, a word on the uncertainties associated with the data would be necessary. Not clear to me is the distinction between the data serving for the development of MedOC4 and the data used for validation with match-ups (points of the former going into the latter are not fully independent validation points). As they are used for validation, BOUSSOLE and AERONET-OC data should be described a bit better. While I'm not familiar with the BOUSSOLE data distribution, there is a clear data policy for use of the AERONET-OC data (offer of authorship if I'm not mistaken) and I'm wondering if this has been respected (there is not even an acknowledgment in the manuscript).

*Figure 1 showing the Cal/Val dataset will be redrawn to better show their distribution, and from which it will appear clear that the calibration data are not being used for validation. The reviewer is right; the section acknowledgement was missing. We will fill it acknowledging AERONET-OC, BOUSSOLE, BGC-Argo and NASA OBPG as those providing high quality data and which we are grateful to. The AERONET-OC PI declined our offer of co- authorship .*

Below are detailed comments, with requests for clarification/corrections and suggestions for improving the text. I'd recommend numbering the sections and sub-sections.

*We will number all sections and sub-sections.*

l.10: I'd suggest: "multi-sensor processing applied to the Mediterranean Sea by the Ocean Colour Thematic Assembly Centre of the Copernicus. . .": The abstract should be readable by readers who don't know about CMEMS, TAC, . . .

*OK*

l.11: "A basin-scale..."

*OK*

l.12: "to fine-tune"

*OK*

l.14: "than those"

*OK*

l.15: "The Mediteranean...": information associated with this sentence should be relocated in the beginning of the abstract.

*OK*

l.21: "CMEMS delivers..." rather than 'includes'

*OK*

l.27: "users who"

*OK*

l.28: please define all acronyms at first use.

*OK*

l.2: "near-real time (NRT) and delayed time (DT) modes."

*OK*

l.3: in OC jargon, monthly data computed from daily data are still termed L3.

*We will add the terms "Within CMEMS" at the beginning of the sentence to stress that this is the current terminology adopted by the Copernicus service.*

l.15: is there a reference for this approach?

*We will remove the statement from the introduction and add sentence framing this approach in the conclusions.*

l.17-18: heavy sentence about 2 important benefits; should be reworded.

*We will rephrase and split the sentence.*

l.20: "are derived"

*OK*

l.23: "is foreseen": so it is still not the case, which is at variance with the abstract.

*This sentence refers to OLCI, which is still not included in the multi-sensor product. It will probably be so by 2019. We do not see any discrepancy with what is stated into the Abstract, where OLCI is not mentioned.*

l.26: "DT data"

*OK*

l.26: "precise": what does this indicate exactly? l.27: "both to be accurate": does it mean DT and REP? then the sentence should be restructured. What does 'consistent' mean here?

*The sentence will be rephrased into "As such, DT data are expected to be as accurate as timeliness allows. The accuracy of REP data need to be stable in time as these data are consistently processed with a single software version."*

l.27: "For the sake of timeliness. . ."

*OK*

l.4: "This work. . .": this paragraph describes the structure of the work, but the primary objective of the whole study is not clear. Page 2, line 28, "one of the aims" is mentioned, but how does it fit here, and what are the other aims?

*We will add a sentence explicitly stating that the overall objective of the work is to provide Copernicus users with a comprehensive description of the method currently applied in the OCTAC context of CMEMS to produce the multi-sensor ocean colour product over the Mediterranean Sea. Propagating the REP configuration to the DT processing mode is part of the method.*

l.14: "relies on an in situ. . ."

*OK*

l.18: odd sentence; it means "absorption due to CDOM, absorption due to algal and non-algal particles, absorption due to TSM, and both AOPs and IOPs." absorption is part of the IOPs, so which are the others measured? l.20: are the in-situ IOPs used in this study?

*We will rephrase the sentence to avoid confusion and to more clearly specify what are the in situ data that were actually used in this study.*

l.30: are the authors sure about the choice of acronym . . . a large part of the scientific community is sufficiently well-versed in latin languages to understand the meaning of the word.

*We will change to "Multi-level data processing is achieved using the Software for the Elaboration of Radiometer Data Acquisitions (SERDA)".*

l.2: "normalised by"

*OK*

l.3: Kl and Ku should be defined.

*OK*

l.9: "using the primary sub-surface quantities, it is then"

*OK*

l.10: "such as the Q-factor"

*OK*

l.18: "Chl": are Chl data from BOUSSOLE used in this work?

*Actually they are not. We will remove Chl from the sentence.*

l.6: "using the OC-CCI. . ."

*OK*

l.7: this sentence reads: "OC-CCI... at 1-km ... rather than at 4-km for OC-CCI": unclear.

*We will rephrase the entire paragraph.*

l.14: "Single-sensor pre-processing for NRT/DT modes" I presume.

*It will be changed to "NRT/DT single-sensor pre-processing"*

l.15: "quality-checked": what does it entail?

*We will add a sentence to briefly explain the kind of data quality checks that are operationally performed.*

l.15: I assume that the atmospheric correction applied in these cases is l2gen. This should be mentioned together with an appropriate reference.

*We will add a sentence in the previous paragraph to mention that the NRT/DT data are downloaded from the OBPG at NASA that uses the l2gen in its default parameterisation for the L1 to L2 processing. Later in the paragraph we will add similar details for the REP processing.*

l.22: "that reflects". . . "stripes originate"

*OK*

l.32: "the dimension . . .": it happens also for the other sensors, doesn't it?

*Yes. We will rephrase the paragraph to make it clear.*

l.2: "these missing values": what is the benefit of this step?

*We will specify that the benefit is particularly evident in view of the sensor merging.*

l.6: and what about SeaWiFS and MERIS?

*To generalize we removed the reference to MODIS-AQUA and VIIRS. Please see also response to comment P7.L7*

l.6: "space agencies": in that case it is only NASA.

*OK*

l.7: "atmospheric correction failure": this is a strong step; I would assume that Rrs associated with that flag is just not usable as the AC failed according to the software. What is the criterion used by the software to consider the AC a 'failure'? what is then the status of Rrs? The authors should also support this decision with robust evidence that the resulting Rrs is actually valid.

*We will add details on the processing of pixels identified through the bowtie removal flag. We will clarify that the atmospheric correction failure flag is not applied to VIIRS because it overlaps with the bowtie removal flag for almost all water pixels. We tested the 645 granules (each of 3200x3232 pixels) acquired over the Mediterranean Sea in 100 days (10 April 2018 to 18 July 2018) and found that only in 31 pixels the atmospheric correction failure flag was raised for pixels not affected by bow tie deletion or any of the other OBPG standard flags.*

l.8: "to avoid"

*OK*

l.8: "salt and pepper": please define what this refers to.

*OK*

l.9: "removing all isolated pixels...": in general, this part needs clearer explanation. How is an "isolated pixel" defined?

*We will add a sentence to specify the meaning of the isolated pixel.*

l.13: please provide characteristics of the map (extent, resolution).

*OK*

l.17: "spectra" from different missions?

*Yes. We actually wrote "to merge single-sensor Rrs spectra into a single spectrum". Adding from "different missions" would be redundant.*

l.21: "differ by"

*OK*

l.25: "In general": may be removed.

*OK*

l.30: "in theory": and in practice? Might be removed.

*OK*

l.31: "apply algorithms to derive geophysical products"

*OK*

l.3-5: not clear what the idea is here.

*We will remove this sentence.*

l.7: "Differences between MODIS and VIIRS"; and what about SeaWiFS and MERIS? In general some aspects of the manuscript associated with NRT/DT seem focused exclusively on MODIS/VIIRS (that has some logic since they are still active) but how SeaWiFS and MERIS are handled should also be described as the related products are used in the validation analysis.

*We will add a sentence in section 2.2 (satellite data processing chain) to clarify that the current NRT/DT processing only involves MODIS-AQUA and VIIRS platforms, and that for the sole scope of the product validation we used the Multi-sensor production chain described in that section (2.2) to process the entire satellite data archive from 1997, thus including SeaWiFS and MERIS data.*

l.11: "normalized": does the BRDF correction make use of the OCI Chla value? In that, it is not consistent with the MedOC4 values. This should be acknowledged.

*We will add a sentence acknowledging the issue in the result section when commenting the Rrs matchup analysis, focusing on the trade-offs between accuracy and timeliness in the operational oceanography framework. Furthermore, this issue will be also the topic of a new paragraph discussing the the accuracy of Rrs in section 4.*

l.17: "we tested": by doing what? l.19: "inter-calibration": it is certainly a factor but it is not the only one that could apply. I would still argue that geometry can play a significant role in the differences. Operating the atmospheric correction with different bands might also have an impact through the AC code (eliciting different responses by the AC code and its assumptions/simplifications).

*At pixel scale, the cosine of the scattering angle was compared with the ratio between MODIS-AQUA and VIIRS Rrs for each band. As it will be shown by the new figure in Supplementary Material, no specific pattern was evident through this analysis. We will acknowledge that there might be some other factors linked to the AC influencing the difference between sensors.*

l.22: it does not seem that the bias correction operated by CCI is described in that reference.

*OK, reference will be corrected. This paragraph will be partially rewritten to address the series of comments below about the implemented bias-correction method.*

> l.25: "climatological": what are the periods used for each sensor? is it the same? l.28: "two steps": but then 3 are listed. l.29: "temporary" ? Page 8: l.1: aren't the same equations used by CCI? l.5: "weight w" l.8: "and time": not necessary. l.8: "as for OC-CCI": this should be written before. I think CCI also operates some type of spatial averaging.

l.12: "deemed insufficient": on the basis of what?

*One of the output of Figure 6 is that the performances of SeaWiFS at 670 nm were the worst as compared with the other bands and sensors. This justifies the choice of not using this sensor band as reference for bias adjusting the other sensors. This sentence will be rephrased.*

l.13: "40%": what is the reason for so many missing values at 670 nm? Negative values? But earlier in the text, it is written that the quality check removes all incomplete spectra (if Rrs is <0 at any one band). So at that point, working with incomplete spectra for match-ups is inconsistent with the processing chain.

*In the "Flagging & Mosaicking" paragraph, we will correct "one negative value within the spectrum (excluding the NIR bands) is enough for the entire spectrum to be rejected" replacing NIR with 670nm.*

l.22: which "method"?

*We will replace "method" with "merging".*

l.24: "are only the result of the merging procedure"

*OK*

l.26-29: I have to say that I don't understand how the approach works. Volpe et al. (2018) not being published, the explanation should be clearer. For instance, is the climatology field computed as in the previous section? And then it is not clear how the smoothing operates.

*We will rephrase the sentence into "To prevent the occurrence of such horizontal discontinuities, here we apply the smoothing procedure described in Volpe et al. (2018) and based on the use of the climatology field, described below". Volpe et al. (2018) is now published. We will also add a few sentences to better explain how the smoothing operates.*

l.2-8: The examples are not so obvious to me. I don't even see the Rhone plume on any map (should be a pattern leaving the coast. . .).

*We will add labels on Figure 3e to easy the reading.*

l.12: "In both cases": this is true for the cases shown but is it a general result? In any case, there is no reason to expect that the bias-corrected merged data would be closer to the field data than a simple average.

*The validation results of Figure 6 show that this is a general result.*

l.15: "The climatology field is obtained. . .": climatology is also mentioned in the 2 previous sections (bias correction, merging), so it is not clear what this SeaWiFS Chla climatology is for, nor why it is computed in a different way.

*We will make a clear difference between the daily climatology bias map used in the context of the bias correction and the field climatology used for the sensor merging. We will also remove the incorrect bit "using the MedOC4 regional algorithm for CHL (Volpe et al., 2007)". We will add a sentence to mention that the next version of the processing chain will include a climatology field*

*computed following the method adopted in the bias correction section.*

l.20: "this has been estimated": how? (or where?)

*We will add the appropriate reference.*

l.22: "To overcome. . .": but this type of filtering was already introduced for the Level-2 data. Do outliers appear again later in the processing ? Speaking of 'biases' here is not appropriate.

*Since it is mentioned earlier in the manuscript, we will remove this part from this context.*

l.32:"Even though the latter now show performance comparable to that of empirical algorithms. . ."

*OK*

l.1: "in discussing the characteristics that data must have to be used. . .."

*OK*

l.2: "pointed out that . . ..": the sentence should be re-written.

*We will rephrase the sentence*

l.2: "theoretical": to be replaced by 'semi-analytical' (what is a theoretical algorithm?)

*OK*

l.10: "weather conditions"

*OK*

l.15: "To identify. . . the identification. . .": sentence to be re-written.

*We will rephrase the sentence.*

l.17: "average spectra": the covariance matrix is also needed for such an approach.

*We will add the necessary details to the sentence.*

l.21: "user survey": refers to the CCI user consultation?

*Yes, we will change it.*

l.6-24: in general I think this paragraph may be too long as the approach has already been well described in literature. Regardless of length I don't find it clear for a reader without a prior knowledge of the method.

*We hope that after the suggested changes the paragraph will read better.*

l.12: "unable to perform"

*We will add the results of validation analysis performed using the BGC-Argo float L3 data (Organelli et al., 2017, Earth Syst. Sci. Data).*

l.15: "Inherent Optical Properties": I was therefore expecting validation results for the IOPs while there are none.

*We will remove the IOPs from this sentence. A reference to the Pitarch et 2016 al for the assessment of QAA based bbp in the the Mediterranean Sea will be added in the methods section at 2.2.1 where the use of QAA is first described.*

l.18: "in most instances": some example stats would be appropriate.

*We will rephrase this sentence to show that the overall validation results are not significantly affected by the temporal window. We will add a new figure in Supplementary Material to show the statistics behaviour with a variable temporal window.*

Page 12:

l.2: "used the NRT": meaning data processed as they were in NRT conditions (with preliminary ancillary data, calibration known at the time of acquisition, preliminary climatology, etc. . .)?

*We will correct NRT into NRT/DT. As mentioned earlier, we will add a couple of sentences in Section 2.2 (Satellite Data Processing Chain) where we better explain the differences between NRT and DT data and that for the sole scope of the product validation we used the NRT/DT production chain described in that section (2.2) to process the entire satellite data archive from 1997, thus including SeaWiFS and MERIS data.*

l.12: "common spectral behaviour": is this a point that could be discussed?

*We will add some discussion on this point.*

l.18: "significantly": was any statistical test performed?

*No, we did not perform any statistical tests. However, "do not significantly differ" here referred to the fact that the standard deviation bars in Figure 6 do overlap to one another. We will rephrase the statement by replacing "significantly" with "substantially".*

l.4: "significant"; what does it mean here?

*We will change "significant" to "relevant".*

l.5: "In the NIR": in the red?

*OK*

Table 4: why are there less matchups at 670 nm (I understood that only complete spectra were kept)? The fact that the Multi and CCI products have a different number of match-ups should be discussed.

*In the revised version of the manuscript we will expand on this.*

l.12: "NRT": or DT?

*We will correct to NRT/DT.*

Figure 1: not so easy to distinguish validation from development points.

*We will make it better*

Figure 3: remind that the climatology fields are from a daily climatology.

*OK*

Figure 5: "Pope and Fry 1972"?

*Corrected to Mueller (2000).*

Table 5: writes 'REP' versus 'Multi' while Table writes 'CCI' versus 'Multi'; coherence is needed to avoid confusion. While Table 4 shows different numbers of match-ups for the 2 products, here the numbers are the same. Is there an explanation for this?

*A thorough check throughout the manuscript will ensure consistency of the use of REP to identify the dataset and CCIv3 to identify the source processing chain for Rrs.*

Reply to Anonymous Referee #2

*We wish to thank Referee #2 for the positive feedback. We addressed the comments as detailed below with a line-by-line response provided in italic.*

1. It is understood that the current product merging strategy follows the OC-CCI approach. Please explain whether moving to the Copernicus baseline of spectral band selection for the radiometry is planned in the future.

*We will surely comment on this and say that we periodically check upstream data quality by comparison with in situ observations. The outcome of the comparison constitutes the basis for taking any sort of such kinds of decisions. We currently use SeaWiFS as it revealed to be the best sensor in the satellite-in situ data comparison. We will comment on the fact that using a sensor reference means to choose between the need of having the best available quality (e.g., SeaWiFS, in our current case) and to account for the climate-change issue referring to more recent observations (e.g., Copernicus missions).*

2. Future evolution to incorporate Copernicus missions, i.e. S3 OLCI (when ready), should also be brought up and discussed.

*We will comment on this in the next version of the paper. We will mention that, as stated above, we periodically check the impact of ingesting new data sources into the processing chain when they become available. Moreover, it is important to mention that the evolution quality of new sensors to be ingested into the multi-sensor processing chain is also checked in terms of output data quality and in terms of number of observations available to users.*

[revised manuscript text omitted]

**The Mediterranean Ocean Colour Level 3 Operational Multi-Sensor Processing**

Gianluca Volpe[1], Simone Colella[1], Vittorio Brando[1], Vega Forneris[1], Flavio La Padula[1], Annalisa Di Cicco[1], Michela Sammartino[1], Marco Bracaglia[1,2], Florinda Artuso[3], Rosalia Santoleri[1]

[1] Istituto di Scienze Marine, Via Fosso del Cavaliere 100, 00133, Roma, Italy

[2] Università degli Studi di Napoli Parthenope,Via Amm. F. Acton 38, 80133, Naples, Italy

[3] Agenzia nazionale per le nuove tecnologie, l'energia e lo sviluppo economico sostenibile, Dipartimento Ambiente, Centro Ricerche Frascati, Frascati, Italy

*Correspondence to*: Gianluca Volpe (gianluca.volpe@cnr.it)

**Supplementary Material**

[Figure]

**Figure S.1: 2D frequency histogram of the daily Rrs ratio (R, on the Y-axis) and of the difference of the cosine of the scattering angle (δcos) between MODIS-AQUA and VIIRS for each of the six bands of the Multi product (2012-2017). The scattering angle ($\Theta_s$) is defined as $\Theta_s = \frac{180}{\pi}arccos\left(-\cos\left(\frac{\pi}{180}\theta_0\right)\cos\left(\frac{\pi}{180}\theta_S\right) - \sin\left(\frac{\pi}{180}\theta_0\right)\cos\left(\frac{\pi}{180}\varphi_r\right)\right)$, with $\theta_0, \theta_S, \varphi_r$ being the solar and sensor zenith angles and the relative azimuth angle, respectively. R exhibits a substantial variability across the spectrum with the values shown in panels a) and f) presenting the larger differences. Overall, the median values of the Rrs ratios at the six bands are within the range 0.9-1.1. The noticeable feature is the lack of any dependency of R from the geometry of the observation (δcos).**

[Figure]

**Figure S.2: Relative percent difference between Multi and in situ (MedBiOp) Rrs as function of the temporal window used for determining coincidence. Since the satellite pixel overpass time is lost because of the merging procedure, we assume here to be 12 am local time. Superimposed is also the number of matchups. The plot shows that after 3 hours the range of variability is always within 1%, while the number of points used and thus the significance increases.**

| Name | Definition |
|---|---|
| In situ data average | $\overline{X}^M = \dfrac{1}{N}\sum\limits_{i=1}^{N} X_i^M$ |
| Satellite data average | $\overline{X}^E = \dfrac{1}{N}\sum\limits_{i=1}^{N} X_i^E$ |
| Type-2 slope | $S = \dfrac{\sum\limits_{i=1}^{N}\left(X_i^E - \overline{X}^E\right)^2 - \sum\limits_{i=1}^{N}\left(X_i^M - \overline{X}^M\right)^2 + \left[\left\{\sum\limits_{i=1}^{N}\left(X_i^E - \overline{X}^E\right)^2 - \sum\limits_{i=1}^{N}\left(X_i^M - \overline{X}^M\right)^2\right\}^2 + 4\left\{\sum\limits_{i=k}^{N}\left(X_k^E - \overline{X}^E\right)\left(X_k^M - \overline{X}^M\right)\right\}^2\right]^{1/2}}{2\,\sum\limits_{i=k}^{N}\left(X_k^E - \overline{X}^E\right)\left(X_k^M - \overline{X}^M\right)}$ |
| Type-2 intercept | $I = \overline{X}^E - S\cdot\overline{X}^M$ |
| Determination coefficient | $r^2 = \dfrac{\left[\sum\limits_{i=1}^{N}\left(X_i^E - \overline{X}_i^E\right)\left(X_i^M - \overline{X}_i^M\right)\right]^2}{\sum\limits_{i=1}^{N}\left(X_i^E - \overline{X}_i^E\right)^2 \sum\limits_{i=1}^{N}\left(X_i^M - \overline{X}_i^M\right)^2}$ |

| | |
|---|---|
| Root Mean Square Difference | $RMSD = \sqrt{\dfrac{\sum_{i=1}^{N}(X_i^E - X_i^M)^2}{N}}$ |
| Centre-pattern Root Mean Square Difference | $cRMSD = \sqrt{\dfrac{\sum_{i=1}^{N}\left\{\left[X_i^E - \left(\sum_{j=1}^{N} X_j^E\right)\right] - \left[X_i^M - \left(\sum_{k=1}^{N} X_k^M\right)\right]\right\}^2}{N}}$ |
| Bias | $bias = \dfrac{1}{N}\sum_{i=1}^{N}(X_i^E - X_i^M)$ |
| Mean Absolute Error | $MAE = \dfrac{1}{N}\sum_{i=1}^{N}|X_i^E - X_i^M|$ |
| Relative percentage Difference | $RPD = 100 \cdot \dfrac{1}{N}\sum_{i=1}^{N}\dfrac{X_i^E - X_i^M}{X_i^M}$ |
| Absolute percentage Difference | $APD = 100 \cdot \dfrac{1}{N}\sum_{i=1}^{N}\dfrac{|X_i^E - X_i^M|}{X_i^M}$ |

**Table S.1: Metrics used to compare the estimated (satellite-based) dataset XE to a reference, measured in-situ dataset XM. The centre-pattern (or unbiased) Root Mean Square Distance (cRMSD) describes the error of the estimated values with respect to the measured ones, regardless of the average bias between the two distributions.**

| RRS | $\bar{X}^M$ | $\bar{X}^E$ | Slope | Intercept | $r^2$ | RMSD | cRMSD | Bias | MAE | RPD | APD | N |
|---|---|---|---|---|---|---|---|---|---|---|---|---|
| 412 | 0.0059 | 0.0056 | 1.01 | -0.0004 | 0.75 | 0.0013 | 0.0013 | -0.00030 | 0.00098 | -3 | 19 | 155 |
| 443 | 0.0056 | 0.0056 | 0.94 | 0.0003 | 0.80 | 0.0010 | 0.0010 | -0.00004 | 0.00072 | 2 | 15 | 155 |
| 488 | 0.0052 | 0.0050 | 0.96 | 0.0000 | 0.79 | 0.0007 | 0.0007 | -0.00025 | 0.00056 | -3 | 12 | 155 |
| 490 | 0.0052 | 0.0050 | 0.96 | 0.0000 | 0.79 | 0.0007 | 0.0007 | -0.00025 | 0.00056 | -3 | 12 | 155 |
| 510 | 0.0040 | 0.0035 | 1.08 | -0.0009 | 0.79 | 0.0008 | 0.0006 | -0.00057 | 0.00072 | -13 | 19 | 155 |
| 531 | 0.0033 | 0.0029 | 1.05 | -0.0005 | 0.88 | 0.0006 | 0.0005 | -0.00037 | 0.00053 | -11 | 18 | 155 |
| 547 | 0.0027 | 0.0024 | 1.02 | -0.0004 | 0.91 | 0.0005 | 0.0004 | -0.00030 | 0.00044 | -11 | 18 | 155 |
| 555 | 0.0025 | 0.0022 | 1.03 | -0.0004 | 0.92 | 0.0005 | 0.0004 | -0.00028 | 0.00041 | -12 | 18 | 155 |
| 667 | 0.0003 | 0.0003 | 1.57 | -0.0002 | 0.88 | 0.0002 | 0.0002 | -0.00001 | 0.00010 | -3 | 35 | 117 |
| 670 | 0.0003 | 0.0003 | 1.54 | -0.0002 | 0.89 | 0.0002 | 0.0002 | -0.00001 | 0.00009 | -5 | 34 | 127 |

**Table S.2: Statistics associated with the MODIS-AQUA Rrs computed over the MedBiOp dataset.**

| RRS | $\overline{X}^M$ | $\overline{X}^E$ | Slope | Intercept | r2 | RMSD | cRMSD | Bias | MAE | RPD | APD | N |
|---|---|---|---|---|---|---|---|---|---|---|---|---|
| 410 | 0.0049 | 0.0043 | 1.41 | -0.0026 | 0.44 | 0.0011 | 0.0009 | -0.00065 | 0.00092 | -13 | 19 | 93 |
| 412 | 0.0049 | 0.0043 | 1.39 | -0.0026 | 0.45 | 0.0011 | 0.0009 | -0.00065 | 0.00092 | -13 | 19 | 93 |
| 443 | 0.0049 | 0.0048 | 0.97 | 0.0000 | 0.63 | 0.0007 | 0.0007 | -0.00018 | 0.00054 | -3 | 11 | 93 |
| 486 | 0.0052 | 0.0049 | 0.92 | 0.0001 | 0.86 | 0.0006 | 0.0005 | -0.00034 | 0.00049 | -6 | 9 | 93 |
| 490 | 0.0052 | 0.0049 | 0.93 | 0.0000 | 0.87 | 0.0006 | 0.0005 | -0.00037 | 0.00050 | -6 | 9 | 93 |
| 510 | 0.0044 | 0.0037 | 1.02 | -0.0007 | 0.92 | 0.0008 | 0.0005 | -0.00065 | 0.00069 | -15 | 16 | 93 |
| 551 | 0.0031 | 0.0027 | 0.99 | -0.0003 | 0.95 | 0.0006 | 0.0005 | -0.00036 | 0.00045 | -12 | 15 | 93 |
| 555 | 0.0030 | 0.0027 | 1.00 | -0.0003 | 0.95 | 0.0006 | 0.0005 | -0.00035 | 0.00044 | -13 | 15 | 93 |
| 670 | 0.0005 | 0.0005 | 1.14 | -0.0001 | 0.95 | 0.0002 | 0.0002 | -0.00003 | 0.00010 | -11 | 20 | 50 |
| 671 | 0.0005 | 0.0005 | 1.14 | -0.0001 | 0.95 | 0.0002 | 0.0002 | -0.00003 | 0.00010 | -10 | 20 | 49 |

**Table S.3: Statistics associated with the VIIRS Rrs computed over the MedBiOp dataset.**

RRS
… [3]

| RRS | $\overline{X}^M$ | $\overline{X}^E$ | SLOPE | INTERCEPT | $R^2$ | RMSD | CRMSD | BIAS | MAE | RPD | APD | N |
|---|---|---|---|---|---|---|---|---|---|---|---|---|
| 412 | 0.0085 | 0.0082 | 1.04 | -0.0007 | 0.64 | 0.0019 | 0.0018 | -0.00031 | 0.00146 | 1 | 21 | 98 |
| 443 | 0.0078 | 0.0076 | 0.92 | 0.0003 | 0.62 | 0.0016 | 0.0016 | -0.00028 | 0.00120 | 3 | 21 | 98 |
| 490 | 0.0059 | 0.0056 | 0.69 | 0.0015 | 0.32 | 0.0013 | 0.0013 | -0.00036 | 0.00092 | -1 | 17 | 98 |
| 510 | 0.0040 | 0.0034 | 0.15 | 0.0029 | 0.02 | 0.0012 | 0.0011 | -0.00053 | 0.00084 | -8 | 22 | 98 |
| 555 | 0.0019 | 0.0018 | 0.26 | 0.0013 | 0.03 | 0.0007 | 0.0007 | -0.00009 | 0.00039 | 0 | 20 | 98 |
| 670 | 0.0002 | 0.0003 | 11.16 | -0.0017 | 0.08 | 0.0002 | 0.0001 | 0.00011 | 0.00012 | 76 | 80 | 40 |

Table S.4: Statistics associated with the SeaWiFS Rrs computed over the MedBiOp dataset.

| RRS | $\bar{X}^M$ | $\bar{X}^E$ | SLOPE | INTERCEPT | $R^2$ | RMSD | CRMSD | BIAS | MAE | RPD | APD | N | $\bar{X}^M$ |
|---|---|---|---|---|---|---|---|---|---|---|---|---|---|
| 412 | Mrrs412 | 0.0083 | 0.0092 | 0.97 | 0.0011 | 0.90 | 0.0012 | 0.0009 | 0.00089 | 0.00104 | 13 | 14 | 90 |
| 443 | Mrrs443 | 0.0076 | 0.0081 | 0.85 | 0.0016 | 0.86 | 0.0010 | 0.0009 | 0.00046 | 0.00072 | 9 | 12 | 90 |
| 490 | Mrrs490 | 0.0060 | 0.0057 | 0.56 | 0.0023 | 0.79 | 0.0011 | 0.0010 | -0.00031 | 0.00056 | -3 | 8 | 90 |
| 510 | Mrrs510 | 0.0042 | 0.0036 | 0.54 | 0.0014 | 0.87 | 0.0011 | 0.0010 | -0.00051 | 0.00061 | -10 | 13 | 90 |
| 560 | Mrrs555 | 0.0022 | 0.0019 | 0.54 | 0.0008 | 0.95 | 0.0012 | 0.0012 | -0.00022 | 0.00030 | -5 | 10 | 90 |
| 670 | Mrrs670 | 0.0002 | 0.0002 | 1.17 | -0.0001 | 0.74 | 0.0001 | 0.0001 | -0.00003 | 0.00005 | -13 | 28 | 72 |

**Table S.5: Statistics associated with the MERIS Rrs computed over the MedBiOp dataset.**

RRS
... [5]

| RRS | $\bar{X}^M$ | $\bar{X}^E$ | SLOPE | INTERCEPT | $R^2$ | RMSD | CRMSD | BIAS | MAE | RPD | APD | N |
|---|---|---|---|---|---|---|---|---|---|---|---|---|
| 412 | 0.0069 | 0.0060 | 0.97 | -0.0007 | 0.83 | 0.0015 | 0.0012 | -0.00087 | 0.00112 | -12 | 17 | 262 |
| 443 | 0.0066 | 0.0063 | 0.86 | 0.0006 | 0.82 | 0.0011 | 0.0011 | -0.00030 | 0.00077 | -1 | 13 | 262 |
| 490 | 0.0057 | 0.0053 | 0.75 | 0.0010 | 0.72 | 0.0010 | 0.0009 | -0.00040 | 0.00067 | -4 | 12 | 262 |
| 510 | 0.0042 | 0.0037 | 0.80 | 0.0003 | 0.74 | 0.0010 | 0.0009 | -0.00051 | 0.00067 | -9 | 16 | 262 |
| 555 | 0.0025 | 0.0023 | 0.80 | 0.0003 | 0.84 | 0.0008 | 0.0008 | -0.00017 | 0.00038 | -3 | 14 | 262 |
| 670 | 0.0003 | 0.0003 | 0.97 | 0.0000 | 0.87 | 0.0001 | 0.0001 | -0.00001 | 0.00009 | 5 | 38 | 223 |

**Table S.6: Statistics associated with the CCIv3 Rrs computed over the MedBiOp dataset.**

| RRS | $\bar{X}^M$ | $\bar{X}^E$ | SLOPE | INTERCEPT | $R^2$ | RMSD | CRMSD | BIAS | MAE | RPD | APD | N |
|---|---|---|---|---|---|---|---|---|---|---|---|---|
| 412 | 0.0070 | 0.0064 | 0.99 | -0.0006 | 0.77 | 0.0015 | 0.0014 | -0.00060 | 0.00113 | -7 | 18 | 272 |
| 443 | 0.0066 | 0.0064 | 0.86 | 0.0007 | 0.73 | 0.0013 | 0.0013 | -0.00023 | 0.00089 | 1 | 15 | 272 |
| 490 | 0.0057 | 0.0052 | 0.65 | 0.0015 | 0.55 | 0.0013 | 0.0012 | -0.00047 | 0.00077 | -5 | 13 | 272 |
| 510 | 0.0042 | 0.0037 | 0.65 | 0.0009 | 0.57 | 0.0013 | 0.0011 | -0.00060 | 0.00077 | -11 | 18 | 272 |
| 555 | 0.0025 | 0.0022 | 0.68 | 0.0005 | 0.71 | 0.0012 | 0.0012 | -0.00027 | 0.00044 | -6 | 16 | 272 |
| 670 | 0.0003 | 0.0003 | 1.19 | -0.0001 | 0.91 | 0.0002 | 0.0002 | -0.00002 | 0.00008 | -3 | 35 | 197 |

**Table S.7: Statistics associated with the Multi Rrs computed over the MedBiOp dataset.**

RRS                                                    … [7]

| CHL | $\overline{X}^M$ | $\overline{X}^E$ | SLOPE | INTERCEPT | $r^2$ | RMSD | CRMSD | BIAS | MAE | RPD | APD | N |
|---|---|---|---|---|---|---|---|---|---|---|---|---|
| REP | 0.257 | 0.163 | 0.737 | -0.306 | 0.75 | 0.411 | 0.400 | -0.093 | 0.143 | 7 | 47 | 710 |
| MULTI | 0.257 | 0.158 | 0.752 | -0.309 | 0.74 | 0.427 | 0.415 | -0.098 | 0.146 | 3 | 47 | 710 |
| REP$_{AV}$ | 0.335 | 0.271 | 1.052 | -0.108 | 0.57 | 0.207 | 0.197 | -0.064 | 0.148 | -18 | 43 | 44 |
| MULTI$_{AV}$ | 0.335 | 0.278 | 1.184 | -0.047 | 0.50 | 0.271 | 0.265 | -0.057 | 0.176 | -17 | 48 | 44 |

Table S.8: Statistics associated with satellite Chl computed over the MedBiOp dataset. Satellite Chl is the REP (derived by the application of the MedOC4.2018 to the Rrs derived from the CCIv3 processor) and NRT/DT (derived by the application of the MedOC4.2018 to the Rrs derived from the Multi processing). A subset of matchups on the period (2012 to present) in which VIIRS and MODIS-AQUA co-exist (REP$_{AV}$ and Multi$_{AV}$) is also reported.

| KD490 | $\overline{X}^M$ | $\overline{X}^E$ | SLOPE | INTERCEPT | $r^2$ | RMSD | CRMSD | BIAS | MAE | RPD | APD | N |
|---|---|---|---|---|---|---|---|---|---|---|---|---|
| MULTI | 0.053 | 0.072 | 0.775 | -0.165 | 0.75 | 0.048 | 0.044 | 0.019 | 0.02 | 44 | 47 | 420 |
| GLOBAL | 0.053 | 0.058 | 0.857 | -0.145 | 0.83 | 0.023 | 0.023 | 0.005 | 0.01 | 15 | 24 | 420 |

Table S.9: Statistics associated with satellite-derived kd490 computed in correspondence of the BGC-Argo float dataset (Organelli et al., 2017), whose location is shown in Figure 10b of the main text. The two rows refer to the satellite-derived Kd490 obtained with the Medkd.2018 and the Global algorithm, respectively, as shown in Figure 5 of the main text.

| CHL | SLOPE | INTERCEPT | $r^2$ | RMSD(LOG) | CRMSD(LOG) | BIAS(LOG) | N |
|---|---|---|---|---|---|---|---|
| REP | 0.74 | -0.306 | 0.75 | 0.25 | 0.25 | -0.0428 | 710 |
| MULTI | 0.75 | -0.309 | 0.74 | 0.26 | 0.25 | -0.0600 | 710 |

| CAR | 0.72 | -0.056 | 0.76 | 0.31 | 0.31 | 0.0066 | 14582 |

Table S.10: Chl matchup statistics as in Table S.8 for REP and Multi. The difference is that Chl log-transformation is here used to compute all the statistical parameters, not only those associated with the linear fit (Slope, Intercept and determination coefficient). CAR statistics are those of Figure 8 in the Ocean Colour Climate Change Initiative (Phase Two) Climate Assessment Report (http://esa-oceancolour-cci.org/?q=webfm_send/702).

| Page 3: [1] Deleted | Gianluca Volpe | 11/19/18 4:04:00 PM |
| --- | --- | --- |
| Page 4: [2] Deleted | Gianluca Volpe | 11/19/18 4:04:00 PM |
| Page 5: [3] Deleted | Gianluca Volpe | 11/19/18 4:10:00 PM |
| Page 6: [4] Deleted | Gianluca Volpe | 11/19/18 4:13:00 PM |
| Page 7: [5] Deleted | Gianluca Volpe | 11/19/18 4:14:00 PM |
| Page 8: [6] Deleted | Gianluca Volpe | 11/19/18 4:15:00 PM |
| Page 9: [7] Deleted | Gianluca Volpe | 11/19/18 4:16:00 PM |

---

## Author Response (AR2)

Dear Editor,

In this new version of the manuscript we have again addressed all comments and concerns raised by Reviewer #1 to whom we are very grateful. While editing the text we realized that there was an oversight in the formulation of the MedKd.2018 algorithm, which had to do with the fact that we considered the water contribution twice. We did update the algorithm and

5 results are now much more sound as described by the totally new paragraph about the matchup results for Kd490. So once again, thanks to the Reviewer #1 for raising the concern about the unexpected results previously presented.

I would like the authors to further improve on the clarity of the NRT/DT/REP characteristics. The abstract concludes saying that the study "demonstrates that the NRT processing chain compares sufficiently well with the historical in situ datasets to be

- 10 confidently used also for reprocessing the full data series". This should not be interpreted literally, particularly by users of the NRT data; as I understood the text, no validation results are shown for (strictly) NRT data so that the results of the manuscript are not valid for this type of data. This should be clearly stated in appropriate places in the text and the conclusion of the abstract modified accordingly. Again if I understand well, the authors have recently applied their DT processing chain to the full time series for this paper; considering that the NASA missions are then in a R2018 status and the MERIS data associated
- 15 with the latest ESA reprocessing (post 2012), this DT time series is in practice a REP time series at least up to 2018 when the calibration of MODIS and VIIRS might start diverging from the R2018 parameters. Again from a user point of view, the quality of the DT data might become degraded for new data. This should also be simply stated. *Done:*
- 20

25

- The mention of the NRT from the abstract has been removed and replaced with a reference to the title "operational multi-sensor processing". Hence lines 17-18 now read: "Here we demonstrate that the operational multi-sensor processing chain compares sufficiently well with the historical in situ datasets to be confidently used also for reprocessing the full data time series.".
  - 2) Following comment on page 13 l.24 we clarified that the validation analysis is conducted with the processing chain in the DT mode (and not NRT) and the sentence now reads: "we used the NRT/DT production chain described in Section 2.2 to process the entire satellite data archive and hence generating a consistent DT dataset".
- 3) A new sentence was added on the conclusions and reads: "As the accuracy of the L3 DT data depends also on the sensor calibration of the L2 data used in the NRT/DT processing chain, the L3 operational products might become degraded for newer data if the calibration of the sensors starts diverging from the R2018 parameters.".
- 30 The authors should also consider the minor comments that follow to further improve the text.

1.16: "kd": kd and Kd are used randomly in the manuscript. Please stick to one symbol. *Done. We also updated two figures containing "kd".*

1.16: "reflectance"

Done.

5 1.2: remove "on" *Done*.

1.11 : "The next..."

10 Done.

1.11-12: "radiance", "irradiance" (2)

Done.

15 1.25: "locations"

Done.

1.5: "processed with"

Done.

20 1.7: "the only changes being the input data..."

Done.

1.9: "(NASA processing version R2018.0) currently rely on"

Done.

1.10: "L2 data"

25 Done.

1.11: "which uses"

Done.

1.15: "As detailed later"

Done.

30 1.21: "VIIRS data are derived ..."

Done.

1.22: "MODIS-AQUA data"

Done.

1.25: "In this work,"

Done.

1.26: "this chain, Chl and Kd490": I interpret that 'REP' refers to Chl and Kd490 derived from CCI Rrs?

Correct!

1.31: "chain is used ... are obtained" (for tense consistency).

**5 Done.**

1.32: "MERIS data are from" *Done*.

**Page 6**

10 1.24: "removal flag of l2gen"

Done.

**Page 7**

1.6: "nearest-neighbour approach": ? if this is the case, and considering that the missing pixel is surrounded by valid values,

**15 which one is chosen?**

The reviewer is right, as now stated we use the median value of the surrounding valid pixels.

1.10: this is in effect produces a sensor-specific daily L3 product, right? Is simple averaging used?

Yes. We added "by simple averaging" to the text.

1.26: "estimate Rrs": is a MODIS 510 nm band computed? Is there a special treatment as this band is kind of between 488 and

**20 531?**

Done. A few sentences were added to better detail the specific case of the 510 nm band. 1.29: "from which it is then possible to derive..."

Done.

**25 Page 8**

1.10: "The magnitude of the differences" ? An order of magnitude difference would suggest a factor 10.

Done.

1.10: considering that its results are well described in this paragraph, I am wondering if having Fig. S1 in supplementary material is appropriate. It could also be included in the main text.

30 Done.

1.12: "443 nm"

Done.

1.17: "responsible for"

Done.

1.4: "reduce the spatial gradients": but what is described here only relates to time.

Done.

**5 Page 10**

12: "on the use of a SeaWiFS daily climatology field": this would help understanding the method description.

Done.

1.7: "sharp gradients": at that stage between one-sensor daily map and its associated climatology?

Done. We specified that the procedure refers to the prevention of gradients in the final merged product (thus between sensors).

10 1.21: I'd suggest to reword as: "the satellite Rrs benefiting from the bias correction are closer to the in situ measurements at all bands".

Done.

**Page 11**

**15 1.21: "O'Reilly"**

Done.

**Page 12**

1.4 : "For Case I ...": by D'Alimonte et al. ? I don't think they used the MedOC data set. This sentence introduces someconfusion in this paragraph (and might removed).

**Done.**

1.33: "whose functional ...": does this bring some additional information to the sentence? It might just be better to say "ratio between 490 and 555 nm".

**Done.**

**25**

1.6: these are 'grid points' and no longer 'pixels'.

**Done.**

1.8: "10 am": Fig. S2 says 12am.

30 Done. Caption of Fig.S2 has been corrected accordingly.

1.24: "NRT/DT": this should be a place to clarify that the validation analysis is conducted with the processing chain in the DT mode (and not NRT).

Done. The sentence was extended to read: "we used the NRT/DT production chain described in Section 2.2 to process the entire satellite data archive and hence generating a consistent DT dataset"

1.9: "(RPD of 76%)"

Done.

5 1.11: "Table S2-S7": from these tables, it seems that the band-shifted data are also compared with respect to in-situ data; this should be mentioned in the table caption. Similarly, if there are validation statistics for all bands of each mission, it is likely that in-situ data have also been band-shifted when necessary. This is not mentioned anywhere I think. Units should be mentioned.

Done. A symbol has been added to tables S2-S7 where necessary to indicate the band-shifted wavelengths. Moreover, a

10 sentence has been added in section 2.1 (The Mediterranean Sea in situ bio-optical dataset: MedBiOp) to specify that the in situ data spectral resolution was increased via band-shifting to allow the satellite – in situ matchup. A similar sentence was also added to the caption of Table 1. Units were added to the table captions.
1.13: Table 2: units (% or geophysical) should be indicated.

Done.

15 1.31: "larger": in relative terms. APD is largest at 670 nm, but lowest in RMSD. *Done.*

1.3: I'd say: "differences between these two products being smaller than 5%", to avoid confusion and clarify that this is a comparison with respect to in-situ data.

Done.

20

1.4: "R2014 and R2018": yes but note that there are other notable differences between Multi and CCI, such as the use of Polymer.

Done.

25 1.6: "is seen at 670 nm" (to avoid repetition).

Done.

1.11-12: isn't it the same message as lines 1-2?

Done. This sentence has been removed.

1.17: MERIS is also processed with POLYMER, no?

30 Yes it is, but here we are describing possible sources of differences and MERIS is processed with POLYMER in both chains (Multi and CCI).

1.26: "merged.." what? Product?

Done.

1.7: "CRMSD"

Done.

1.9: "global set ... a larger modal..."

5 Done.

1.18: "Figure 6 and Table S7": why should stats for kd be similar with Rrs results? 1.19: it is also surprising that the Med algorithm is actually worse (in terms of bias). 1.20: "one such reason"

As mentioned earlier, we rephrased the entire paragraph on the basis of the new results on the Kd matchup analysis.

10 Page 17

1.3: "CCI, 2016b" ?*Done*.1.6: "merging approach"*Done*.

15 1.7: "locations"

Done.

Fig.1: "in situ stations"

Done.

20 Fig. 3: what do the curves within the color bars mean?

They show the distribution histogram of the image. A sentence has been added to the figure caption.

Fig. 8: "red and orange": isn't it black and blue?

Done.

Fig.9: "NRT": this should be NRT/DT.

25 Done.

Fig.10: "space-time distribution": panel c?

Done.

Table 5: "the three matchup data set": three? Units should be given when appropriate (same for all Tables S).

Done.

30 Table S9: "whose location is shown in Figure 10c"?

Done.

**The Mediterranean Ocean Colour Level 3 Operational Multi-Sensor Processing**

Gianluca Volpe1, Simone Colella1, Vittorio E. Brando1, Vega Forneris1, Flavio La Padula1, Annalisa Di Cicco1, Michela Sammartino1, Marco Bracaglia1,2, Florinda Artuso3, Rosalia Santoleri1

1 Istituto di Scienze Marine, Via Fosso del Cavaliere 100, 00133, Roma, Italy
 2 Università degli Studi di Napoli Parthenope, Via Amm. F. Acton 38, 80133, Napoli, Italy
 3 Agenzia nazionale per le nuove tecnologie, l'energia e lo sviluppo economico sostenibile, Dipartimento Ambiente, Centro Ricerche Frascati, Frascati, Italy

Correspondence to: Gianluca Volpe (gianluca.volpe@cnr.it)

- 10 Abstract. The Mediterranean near-real-time multi-sensor processing chain has been set up and is operational in the framework of the Copernicus Marine Environment Monitoring Service (CMEMS). This work describes the main steps operationally performed to enable single ocean colour sensors to enter the multi-sensor processing applied to the Mediterranean Sea by the Ocean Colour Thematic Assembly Centre within CMEMS. Here, the multi-sensor chain takes care of reducing the inter-sensor bias before data from different sensors are merged together. A basin-scale in situ bio-optical dataset is used both to fine-tune
- 15 the algorithms for the retrieval of phytoplankton chlorophyll and attenuation coefficient of light, Kd, and to assess the uncertainty associated with them. The satellite multi-sensor remote sensing reflectance spectra better agree with the in situ observations than those of the single sensors. Here we demonstrate that the operational multi-sensor processing chain compares sufficiently well with the historical in situ datasets to be confidently used also for reprocessing the full data time series.

**1 Introduction**

- 20 The Copernicus Marine Environment Monitoring Service (CMEMS) is one of the six services of the Copernicus program. It provides regular and systematic reference information on the physical state, variability and dynamics of the ocean, ice and marine ecosystems for the global ocean and the European seas. CMEMS delivers both satellite and in-situ high-level products prepared by Thematic Assembly Centres (TACs) and modelling and data assimilation products prepared by Monitoring and Forecasting Centres (MFCs). The Ocean Colour Thematic Assembly Centre (OCTAC) builds and operates the European ocean
- 25 colour operational service within CMEMS providing global, Pan-European and regional (Arctic Ocean, Atlantic Ocean, Baltic Sea, Black Sea, and Mediterranean Sea) ocean colour (OC) products based on earth observation from OC missions (Le Traon 2015, Von Schuckman 2017). The OCTAC bridges the gap between space agencies, providing ocean colour data, and all users who need the added-value information not available from space agencies. Presently, the OCTAC relies on current and legacy OC sensors: MERIS (MEdium Resolution Imaging Spectrometer) from ESA, SeaWiFS (Sea-viewing Wide Field-of-view
- 30 Sensor) and MODIS (Moderate Resolution Imaging Spectroradiometer) from NASA, VIIRS (Visible Infrared Imager

[revised manuscript text omitted]

- 2) For each day, the ratio between the temporary average Rrs maps from the various sensors is computed.
- This allows the calculation of 365 daily climatology maps of the ratio between each pair of missions over the periods of reference.
- 4) To increase map coverage, smoothing of the daily climatology bias maps over a temporal window of 2N+1 days (with
  - N=60) are computed following equations 1 and 2:

$$\boldsymbol{\delta}(\boldsymbol{d}, \boldsymbol{x}, \boldsymbol{y}) = \frac{\sum_{l=-N}^{N} w_l \delta_l (\boldsymbol{d} + \boldsymbol{i}, \boldsymbol{x}, \boldsymbol{y}) \theta_l}{\sum_{l=-N}^{N} w_l \theta_l}, \
[revised manuscript text omitted]

14

| 1 | Deleted: Figure 6      |    |
|---|------------------------|----|
| 1 | Formatted: Font: 10 pt | ٦. |

| Г | ormati | ea:  | FORU: | 10 | pı, | NOL | DOI |
|---|--------|------|-------|----|-----|-----|-----|
| D | eleted | : Ta | ble 1 |    |     |     |     |

| Deleted: Figure 7 |  |
|-------------------|--|
| Deleted:          |  |
| Deleted: Figure 7 |  |

(MedBiOp and BOUSSOLE) but not in the coastal domain (AAOT). CCIv3 and Multi Rrs present a general good agreement, differences between these two products and the in situ data being smaller than 5%; this low difference is likely due to the two source datasets which are derived from two different NASA reprocessings, R2014 and R2018, and partially to the use of POLYMER for the MODIS-AQUA processing in the CCI chain; at 412 nm, and to a lesser extent at 443 nm, this difference

- 5 is more pronounced (more than 5%) because the impact of the R2018 is larger at these bands. An even more evident difference (larger than 10%) is seen at 670 nm; here the impact of R2018 should be less important than in the blue bands. One important difference between Multi and CCIv3 is that Multi is not bias-corrected over SeaWiFS at this band (section 2.2.2); since SeaWiFS performances at this band are not as good as at the other bands it is reasonable to assume that this might be the cause of the observed discrepancy, further supporting the choice of not using SeaWiFS to bias-correct the other sensors in this band.
- 10 Another important feature in Figure 9, is the general difference of satellite performance (both Multi and CCIv3) in coastal and open waters

Table 4 shows the number of matchups for each band of Multi and CCIv3 in correspondence of the three in situ datasets. Two aspects emerge: one linked to the difference between Multi and CCIv3 and the other between the 670 nm band and the other bands. As mentioned earlier, it is reasonable to assume that the different source data (R2018.0 for Multi and R2014.0 for

- 15 CCIv3) is responsible for the differences in spatial coverage and hence in the number of matchups. Moreover, it should be mentioned that MODIS-AQUA used in the Multi processing chain derives from NASA R2018.0, while it derives from POLYMER atmospheric correction scheme for CCIv3. As for the differences between the 670 nm band and the other bands, the very noisy spatial patterns present in the daily images of the Rrs at 670 nm very often result, at the scale of the matchup pixels, in the coefficient of variation to exceed the 20% threshold (section 2.3).
- 20 Overall, despite their absolute differences, the two multi-sensor satellite products show a similar level of accuracy which suggests that the Multi processor is also suitable for the REP processing chain. This would provide the two benefits of reducing the number of upstream data provider and of giving the NRT/DT and REP products full compatibility. *Matchup – Chl*

Figure 10, shows the matchups for the L3 Chl product for both processing modes, REP (derived from the CCIv3 Rrs) and

- 25 NRT/DT (derived from the Multichain described in this study). To facilitate the comparison between the two satellite products, the matchup dataset includes only the points in which both satellite data are available. Both products are regularly distributed around the line of best agreement for the entire Chl range, although for in situ values larger than 0.3 mg m-3 there is a noticeable dispersion increase. Table 5 shows the statistics of the four datasets plotted in Figure 10. To assess the uncertainties of the Multi Chl currently distributed on the CMEMS portal, the analysis was performed on the period in which VIIRS and MODIS
- 30 co-exist, i.e. January 2012 onwards. Despite the different number of matchups (44 vs 710) and different Chl ranges (~0.04 2 vs ~0.007 9), statistics associated with the full time series are totally comparable with those obtained with the most recent data only (2012 to present) as denoted by the AV (MODIS-AQUA and VIIRS) subscript in both Figure 10 and Table 5. To further assess the level of accuracy associated with the Chl retrieval from multi mission merged approach presented in this study, we compared with the results at global scale reported in Climate Assessment Report, CAR, (CCI, 2017). Differently

15

**Deleted: Figure 8**

| -( | Deleted: Figure 9       |
|----|-------------------------|
| (  | Deleted: multi          |
| (  | Deleted: -sensor merged |
|    |                         |

| Formatted: Font: 10 pt, Not Bold |  |
|----------------------------------|--|
| Deleted: Table 5                 |  |
| Deleted: Figure 9                |  |

| •( | Deleted: Figure 9                |
|----|----------------------------------|
| (  | Formatted: Font: 10 pt, Not Bold |
| (  | Deleted: Table 5                 |

than here, in the CAR, the Chl log-transformation was used to compute all the statistics, not only those associated with the linear fit (slope, intercept and determination coefficient, section 2.3). Therefore, for this analysis, we recomputed all the statistics of Table 5, accordingly (Table S.10). The in situ data used to compute the CAR statistics are much more numerous (14582, Table S.10). Nonetheless, results for the proposed regional algorithms as well as for CCI at global and Mediterranean

- 5 scales show a general good agreement in terms of the determination coefficient, RMSD, CRMSD and the slope of the linear fit. The difference in the intercept only reflects the difference in the two dataset range of variability, with the global set being wider and characterized by a larger modal value (centred over ~1 mg m-3, Figure 8 in CAR) than the MedBiOp (Figure 1Q). Matchup – Kd490
- Figure 11a shows the validation result of the satellite-derived Kd490 with respect to the in situ Kd490 obtained from the BGC-10 Argo float dataset (Organelli et al., 2016), whose space-time distribution is shown in Figure 11c. As a matter of comparison, both algorithms shown in section 2.2.3 and in Figure 6b are presented. MedKd performs better than Global algorithm as also highlighted by their matchup statistics (Table S.9), from which it appears that the regional algorithm presents lower biases (both absolute and percent) than the Global. Similarly to the Global, the MedKd algorithm overestimates in situ values larger than 0.1 m-1, probably due to the lower representativeness of the MedBiOp dataset used to derive the algorithm in this range
- 15 of variability (Figure 6b and Figure 11b). Furthermore, the Global algorithm shows a clear overestimation at the lower end of the range of variability with respect to the in situ data, as well as to the regional algorithm, as shown by the two lines of best fit (slopes are 0.86 and 1 for the Global and the MedKd, respectively, Table S.9). On the contrary, the MedKd algorithm performs well at low values.

A similar analysis from Organelli et al. (2017) shows that satellite data overestimate the BGC-Argo – derived Kd490 for values

20 below 0.1 m-1 (their Figure 11), Here, we show that this still holds when the Global algorithm is used, but that the MedKd algorithm corrects for this overestimation. This analysis, performed over a fully independent dataset, justifies and supports the choice of using the MedKd algorithm for the operational chain in lieu of the Global.

**4 Conclusions**

This work presented the latest achievements in the operational processing chain for ocean colour data stream for the 25 Mediterranean Sea in the context of the European Copernicus Marine Environment Monitoring Service. The development of the multi-sensor merged product builds on the previous version of this chain, which was focused on parallel processing of single sensors (SeaWiFS, MODIS and MERIS, Volpe et al., 2012). The introduction of an operational multi-sensor merged product aims to meet the operational oceanography intrinsic requirement of "One Question One Answer". Three main steps were implemented: band-shifting, inter-sensor bias correction, and the sensor merging. The band-shifting is implemented

30 exactly as in Mélin and Sclep (2015), while the implementation of the inter-sensor bias correction differs from the OC-CCI

technique (CCL 2016b) in the temporal and spatial aggregation scales. The sensor-merging shown in this work is based on the use of the climatology as input to the smoothing procedure as described in Volpe et al. (2018). The output of this processing

| Formatted: Font: 10 pt, Not Bold |  |
|----------------------------------|--|
| Deleted: Table 5                 |  |
| Deleted: M                       |  |

| -(   | Deleted: Figure 9     |
|------|-----------------------|
| -(   | Deleted: kd490 |
| -(1  | Deleted: Figure 10    |
| ·(I  | Deleted: kd490        |
| Ó    | Deleted: kd490        |
| -(1  | Deleted: Figure 5     |
| ) (I | Deleted: also         |

**Deleted: ing**

**Deleted:**

Deleted: Here, since the satellite kd490 are derived from the two algorithms applied to the Multi Rrs the level of accuracy that one can expect should be of the same order of magnitude as the one shown in Figure 6 and Table S.7. However, statistics associated with this matchup analysis show that both algorithms do perform worse than expected (Table S.9), especially the Medkd. There can be a series of reasons for which the two datasets (satellite and in situ) show good or bad agreement. One such a reason could be linked to the different space-time distribution of the calibration and validation datasets, but from the comparison of Figure 1 and Figure 10c it does not appear to be the case. The different data distribution can at least partially explain the poor performances of the Medkd with respect to the BGC-Argo kd490 (Figure 10b). As already mentioned in section 2.2.3, the Global algorithm appearing inadequate to represent the Mediterranean Sea conditions was superseded by the regional Medkd algorithm. The fact that in the validation exercise, the Global algorithm performs better than the Medkd is totally unexpected and, as suggested by Organelli et al. (2017), "strongly warrants for further investigation'

chain is the Rrs spectrum that constitutes the input to all algorithms used to derive the various geophysical products. The Rrs computed with the multi-sensor merging approach shows good agreement when compared with in situ observations not only, with the basin-scale MedBiOp dataset but also with the two fixed locations AAOT and BOUSSOLE time series. As the accuracy of the L3 DT data presented in this study depends also on the sensor calibration of the L2 data used in the NRT/DT

5 processing chain, the L3 operational products might become degraded for newer data if the calibration of the sensors starts

diverging from the R2018 parameters. Moreover, this work presents an updated version of the empirical algorithms for Chl and Kd retrievals for the Mediterranean Sea based on the extended MedBiOp dataset. The comparison with the in situ observations yields good results when applied to both the Rrs derived from the CCIv3 processor and those derived from the multi-sensor merged processing shown here. This suggests the opportunity to use the proposed multi-sensor processing chain

10 in the REP context as well.

**5 Acknowledgements**

We wish to thank the anonymous reviewers for detailed and pertinent comments that helped to greatly improve the manuscript. Giuseppe Zibordi, as PI of the AERONET-OC site of Venise, is warmly thanked for the Level-2 surface reflectance data processing and site maintenance. The authors are also grateful to the BOUSSOLE project for maintaining and providing high-

[revised manuscript text omitted]

**7 Figures**

| (  | Formatted: English (US) |
|----|-------------------------|
| -( | Field Code Changed      |
| (  | Formatted: English (US) |
| Ľ  | Formatted: English (US) |
| (  | Formatted: English (US) |
|    |                         |

Figure 1: Study Area and space-time distribution of the in situ MedBiOp dataset (1997-2016) used in this work. Dots identify the in situ stations used as sea-truth for satellite data validation, whereas crosses refer to the observations used to develop the regional OC algorithms.